# MCPH1 inhibits Condensin II during interphase by regulating its SMC2-Kleisin interface

Martin Houlard[1†], Erin E Cutts[2†], Muhammad S Shamim[3,4,5], Jonathan Godwin[1], David Weisz[3,5], Aviva Presser Aiden[3,5], Erez Lieberman Aiden[3,5], Lothar Schermelleh[1], Alessandro Vannini[2,6*], Kim Nasmyth[1*]

[1]Department of Biochemistry, University of Oxford, Oxford, United Kingdom; [2]Division of Structural Biology, The Institute of Cancer Research, London, United Kingdom; [3]The Center for Genome Architecture, Department of Molecular and Human Genetics, Baylor College of Medicine, Houston, United States; [4]Medical Scientist Training Program, Baylor College of Medicine, Department of Bioengineering, Rice University, Houston, United States; [5]Center for Theoretical Biological Physics, Rice University, Houston, United States; [6]Human Technopole, Milan, Italy

*For correspondence:
alessandro.vannini@fht.org (AV);
ashley.nasmyth@bioch.ox.ac.
uk (KN)

[†]These authors contributed
equally to this work

Competing interest: The authors
declare that no competing
interests exist.

Reviewing Editor: Adèle
L Marston, University of
Edinburgh, United Kingdom

**Abstract** Dramatic change in chromosomal DNA morphology between interphase and mitosis is a defining features of the eukaryotic cell cycle. Two types of enzymes, namely cohesin and condensin confer the topology of chromosomal DNA by extruding DNA loops. While condensin normally configures chromosomes exclusively during mitosis, cohesin does so during interphase. The processivity of cohesin's loop extrusion during interphase is limited by a regulatory factor called WAPL, which induces cohesin to dissociate from chromosomes via a mechanism that requires dissociation of its kleisin from the neck of SMC3. We show here that a related mechanism may be responsible for blocking condensin II from acting during interphase. Cells derived from patients affected by microcephaly caused by mutations in the MCPH1 gene undergo premature chromosome condensation. We show that deletion of *Mcph1* in mouse embryonic stem cells unleashes an activity of condensin II that triggers formation of compact chromosomes in G1 and G2 phases, accompanied by enhanced mixing of A and B chromatin compartments, and this occurs even in the absence of CDK1 activity. Crucially, inhibition of condensin II by MCPH1 depends on the binding of a short linear motif within MCPH1 to condensin II's NCAPG2 subunit. MCPH1's ability to block condensin II's association with chromatin is abrogated by the fusion of SMC2 with NCAPH2, hence may work by a mechanism similar to cohesin. Remarkably, in the absence of both WAPL and MCPH1, cohesin and condensin II transform chromosomal DNAs of G2 cells into chromosomes with a solenoidal axis.

## Editor's evaluation

This paper will be of broad interest to scientists in the field of chromosome biology and has high clinical relevance. It reveals how the MCPH1 protein, that is frequently mutated in disorders of brain growth, prevents premature chromosome condensation. Diverse and complementary approaches are used to provide a compelling argument that provides insight into the mechanism by which chromosome organisation is temporally controlled.

## Introduction

The segregation of sister DNAs during mitosis requires that they are first disentangled and organised into individual sister chromatids before being pulled in opposite directions by the mitotic spindle upon the dissolution of sister chromatid cohesion by separase. Chromatid formation during mitosis and sister chromatid cohesion necessary for bi-orientation are mediated by highly related structural maintenance of chromosomes (SMC)-kleisin complexes, namely condensin and cohesin respectively. In both cases, the activity of their ring-like SMC-kleisin trimers is regulated by large hook-shaped proteins composed of tandem HEAT repeats, known as HAWKs (HEAT repeat proteins associated with kleisins) (*Yatskevich et al., 2019*).

The conundrum that complexes with such similar geometries appear to perform such dissimilar functions has been resolved by the realisation that cohesin also organises DNAs into chromatid-like structures, albeit during interphase and only upon ablation of a regulatory protein called WAPL (*Tedeschi et al., 2013*) as well as naturally during meiotic prophase (*Challa et al., 2018*) or during V(D)J recombination when WAPL is downregulated in pro-B cells (*Hill et al., 2020*). There is now considerable evidence that cohesin and condensin organise chromosomal DNAs during interphase and mitosis, respectively, by extruding DNA loops in a processive manner. Indeed, both exhibit such loop extrusion (LE) activity in vitro (*Davidson et al., 2019*; *Ganji et al., 2018*). LE mediated by cohesin is halted or at least retarded at specific sequences bound by CTCF, and the processivity of the complex is reduced by WAPL which induces cohesin's dissociation from chromatin, albeit only infrequently every 10–20 min (*Gerlich et al., 2006*; *Hansen et al., 2017*; *Wutz et al., 2020*). No such site-specific DNA binding proteins are known to regulate condensin. In both cases, chromatid formation is envisaged to arise from processive LE activity, which organises DNA into a series of loops, each emanating from a central core containing the SMC-kleisin loop extruders (*Yatskevich et al., 2019*).

Mammalian cells possess two types of condensin complexes: condensin I and II. Both complexes share the same SMC proteins, SMC2 and SMC4, which both contain 50 nm long anti-parallel coiled coils connecting a hinge domain at one end with an ATPase domain, formed from the N- and C-terminal extremities, at the other. Dimerisation via their hinges creates V-shaped SMC2/4 heterodimers whose ATPase heads are inter-connected by their kleisin subunits, NCAPH1 for condensin I and NCAPH2 for condensin II. These in turn recruit different HAWK regulatory subunits, NCAPD2 or D3 and NCAPG or G2 for condensin I and II, respectively (*Cutts and Vannini, 2020*). While condensin II remains in the nucleus during interphase and starts to organise chromosomal DNAs during prophase, condensin I only has access to the DNA after nuclear envelope break down (*Hirota et al., 2004*; *Ono et al., 2004*; *Ono et al., 2003*). Despite this difference, both accumulate along the longitudinal axes of metaphase chromosomes (*Ono et al., 2004*; *Walther et al., 2018*). HiC data suggest that condensin I extrudes shorter loops that are nested within longer ones previously created by condensin II. Interactions between distant loops suggest that the loops created by condensin II may be organised radially around a coiled or solenoidal axis (*Gibcus et al., 2018*). The ratio between condensin I and condensin II adjusts the coiling of chromosome axes, partly by altering the width of the central spiral and generating curled chromosomes (*Baxter and Aragón, 2012*).

An important factor in limiting the processivity of loop extrusion by cohesin is its release from chromatin. This process that depends on WAPL involves engagement of cohesin's ATPase heads (*Elbatsh et al., 2016*) and dissociation of the N-terminal domain (NTD) of its SCC1 kleisin subunit from the coiled coil (neck) that emerges from SMC3's ATPase head domain (*Beckouët et al., 2016*). Because a co-translational fusion of SCC1's NTD to the C-terminal domain of SMC3's ATPase blocks release, it has been proposed that it involves passage of DNAs previously entrapped inside its SMC-kleisin ring through an exit gate created by dissociation of SCC1 from SMC3 (*Chan et al., 2012*; *Gligoris et al., 2014*). Indeed, the kleisin NTDs appear to dissociate in vitro from engaged SMC2 and SMC4 heads (*Hassler et al., 2019*; *Lee et al., 2020*) as well as those of SMC1 and SMC3 (*Buheitel and Stemmann, 2013*; *Chan et al., 2012*; *Herzog et al., 2014*). This observation raises the possibility that release via a kleisin-SMC exit gate might be a feature of chromosomal condensin complexes as well as cohesin. However, whether release of this nature occurs in vivo and whether it has an important function is not known.

Condensin II's activity and functions within the nuclei of interphase cells are poorly understood. While some studies have reported that it binds to chromatin during interphase (*Ono et al., 2013*), its depletion from post mitotic mouse liver cells has little or no effect on genome organisation or

transcription (*Schwarzer et al., 2017*). Likewise, the notion that condensin II is activated as cells enter mitosis via kinase cascade involving CDK1 and PLK1 (*Abe et al., 2011*) also begs the question as to whether it has any significant role during interphase. However, an important clue that condensin II can in principle function during this stage of the cell cycle comes from the characterisation of patients carrying a mutation of the *Mcph1* gene, which causes a reduction in the size of the cerebral cortex known as primary microcephaly (OMIM 608585) (*Thornton and Woods, 2009*; *Woods et al., 2005*). Cells from such patients have an increased number of prophase-like cells (*Neitzel et al., 2002*; *Trimborn et al., 2004*). The prophase-like organisation of chromosomal DNA has since been shown to be mediated by condensin II (*Trimborn et al., 2006*; *Yamashita et al., 2011*), whose abnormal activity triggers premature condensation in G2 and late de-condensation in G1 (*Arroyo et al., 2019*; *Neitzel et al., 2002*; *Trimborn et al., 2004*).

MCPH1 is only found in metazoa. It contains three BRCT domains, one N-terminal and two C-terminal, separated by a disordered region. The diverse phenotypes of mutant cells have led to many different and often conflicting suggestions for its roles. Binding of MCPH1's C-terminal BRCT domains to phosphorylated histone H2AX (*Venkatesh and Suresh, 2014*) is thought to play a part in the DNA damage response, while the premature chromosome condensation caused by mutations within its N-terminal BRCT has been attributed to this domain's defective association with the SWI/SNF nucleosome remodelling complex and SET1 (*Leung et al., 2011*). MCPH1 is also thought to bind DNA and has been proposed to function in telomere maintenance, centriole organisation and CHK1 activation (*Alderton et al., 2006*; *Chang et al., 2020*; *Cicconi et al., 2020*; *Gruber et al., 2011*; *Lin and Elledge, 2003*). In contradiction to the finding that MCPH1 binds directly to condensin II, it is widely believed that the N-terminal ~200 residues of MCPH1 compete with condensin II for chromosomal binding (*Yamashita et al., 2011*), in other words, MCPH1 has been proposed to occupy loci required for condensin II's chromosomal activity.

Given the diversity and conflicting views of MCPH1's function, we have re-addressed its role in regulating chromosome structure by analysing the consequences of deleting its gene in mouse ES cells. We show that loss of MCPH1 induces a prophase-like organisation of chromatin during G1 and G2 but not during S phase, and that this depends on condensin II. Crucially, this phenotype, which is accompanied by condensin II's stable association with chromosomes, is unaffected by inhibiting CDK1, suggesting that MCPH1 does not inhibit chromosome condensation merely by delaying the CDK1 activation normally necessary for condensin II activity as previously suggested (*Alderton et al., 2006*; *Gruber et al., 2011*; *Tibelius et al., 2009*). We demonstrate that MCPH1 instead regulates the organisation of chromosomal DNA through the binding of condensin II's NCAPG2 subunit by a conserved short linear motif (SLiM) situated within its central domain. Such binding does not per se explain how MCPH1 regulates condensin II and has little or no effect on its ATPase activity, at least in vitro. Presumably, other domains, for example its N-terminal BRCT, which is frequently mutated in human microcephaly patients, have an effector function, by interacting with other sites within the condensin II pentamer or recruit other cellular factors.

A clue to the role of MCPH1 is the resemblance between the stable association of condensin II with chromosomal DNA induced by MCPH1 ablation with the effect on cohesin of mutating *Wapl*, which binds to the cohesin HAWK equivalent to NCAPG2, namely STAG2, also using an FY SLiM. We therefore tested whether like WAPL, MCPH1 prevents condensin II from associating stably with DNA by opening the interface between NCAPH2's NTD with the neck of SMC2's ATPase domain. Remarkably, a translational fusion between SMC2's C-terminus and NCAPH2's N-terminus is not only functional in mouse oocytes but also resistant to the inhibition mediated by an excess of MCPH1 induced by mRNA micro-injection. This raises the possibility that like WAPL, MCPH1 acts by opening the interface between the kleisin's NTD and the neck of the SMC ATPase domain. Finally, we investigated the consequences of inducing the stable association with G2 chromosomes of both cohesin and condensin and found that the chromosomal axes created by cohesin upon mutating *Wapl* are turned into solenoids in cells deleted for both *Wapl* and *Mcph1*.

# Results

## *Mcph1* deletion induces chromosome condensation during interphase in embryonic stem cells

Premature chromosome condensation is a key feature of cells isolated from patients carrying *MCPH1* mutations. To study this in greater detail, we used CRISPR/Cas9 to delete *Mcph1* in mouse E14 embryonic stem cells in which we had previously introduced a Halo-tag at the C-terminal end of NCAPH2 (NCAPH2-Halo) at its endogenous locus. Western blot using an anti-NCAPH2 antibody confirmed that all the NCAPH2 protein present in the cell is shifted to a higher molecular weight that matches the size of the band detected by in-gel Halo-TMR fluorescence. The protein expression levels are identical to the untagged protein in control cells (*Figure 1A*). Deleting the second exon of *Mcph1* induces a frameshift between exon 1 and 3 and thereby complete inactivation of the gene. Western blot analysis of the targeted cells confirmed the lack of any MCPH1 protein (*Figure 1A*).

Immunofluorescence microscopy revealed that *Mcph1* deletion leads to a substantial increase in the fraction of cells with prophase like chromosomes: WT: 6.4%, 12/188 cells compared to *ΔMcph1*: 39.1%, 70/179 cells (*Figure 1B*). FACS analysis of cells stained with propidium iodide to measure DNA content, H3PS10 specific antibodies to detect G2/M cells, and pulse labelled with EdU to identify S phase cells showed no overall change in cell cycle progression (*Figure 1—figure supplement 1*). Thus, *Mcph1* deletion caused little or no prophase arrest. Crucially, all EdU-negative cells, whether they were in G1 (H3PS10 negative, 200 cells counted) or G2 (H3PS10 positive, 200 cells counted), contained prophase-like chromosomes while no condensation was observed in EdU positive S phase cells (200 cells counted) (*Figure 1C* and 3D). In wild-type G1 and G2 cells, different centromeres and peri-centric regions cluster in chromocenters. This feature is abolished in the mutant cells, where every centromere occupies an individual location (*Figure 1—figure supplement 2*) and each chromosome is likewise individualised into prophase-like chromatids (*Figure 1B*). The localisation of NCAPH2 was analysed by labelling its Halo-tag with the fluorescent TMR ligand. In wild type, NCAPH2 has a diffuse nuclear distribution throughout most of the interphase, with no enrichment at any particular site. The protein starts to accumulate at centromeres during G2 and subsequently along the length of chromosomes from prophase until the end of telophase. Deletion of *Mcph1* caused NCAPH2 to associate with the prophase-like chromosomes in all G1 and G2 cells and become enriched at their individualised centromeres (*Figure 1B*).

## MCPH1 restricts Condensin II activity during G1 and G2

The change in chromosome organisation caused by deletion of *Mcph1* is accompanied by a change in the localisation of condensin II. To address whether condensin II is causing the observed phenotype, we used a Halo-PROTAC ligand to specifically induce the degradation of NCAPH2-Halo (*Figure 1D*). Because this completely reversed the re-organisation of chromosomal DNA caused by *Mcph1* deletion (no condensation observed in 150 cells counted, *Figure 1E*), we conclude that altered regulation of condensin II is largely if not completely responsible. Unlike the centromere dispersion, chromosome condensation, and chromosome unpairing induced by loss of Slimb ubiquitin ligase or casein kinase one in *Drosophila* cells, which is accompanied and caused by an increase in the level of NCAPH2 (*Buster et al., 2013*; *Nguyen et al., 2015*), deletion of *Mcph1* in ES cells is accompanied by a modest but nevertheless readily detectable reduction in NCAPH2 levels (*Figure 1A & D*). The implication is that MCPH1 restricts the activity of individual condensin II complexes in G1 and G2 cells.

## MCPH1 prevents Condensin II's stable association with interphase chromatin

Photobleaching studies with GFP-tagged condensin II subunits has revealed that they are highly mobile during interphase, suggesting that they are rarely or only transiently associated with chromatin fibres (*Gerlich et al., 2006*). FRAP of TMR-labelled NCAPH2-Halo confirmed this as photobleached spots of Halo-TMR fluorescence recovered 95 % of their fluorescence within 1 min (*Figure 2*). However, deletion of *Mcph1* caused a dramatic change in the dynamics, with fluorescence merely recovering to 28 % of its starting level within 1 min. After that, no further change occurred during the next 10 min (*Figure 2*), implying that *Mcph1* deletion causes 72 % of condensin II complexes to associate stably with chromatin in G1 or G2 cells, thereby altering chromatin compaction during interphase.

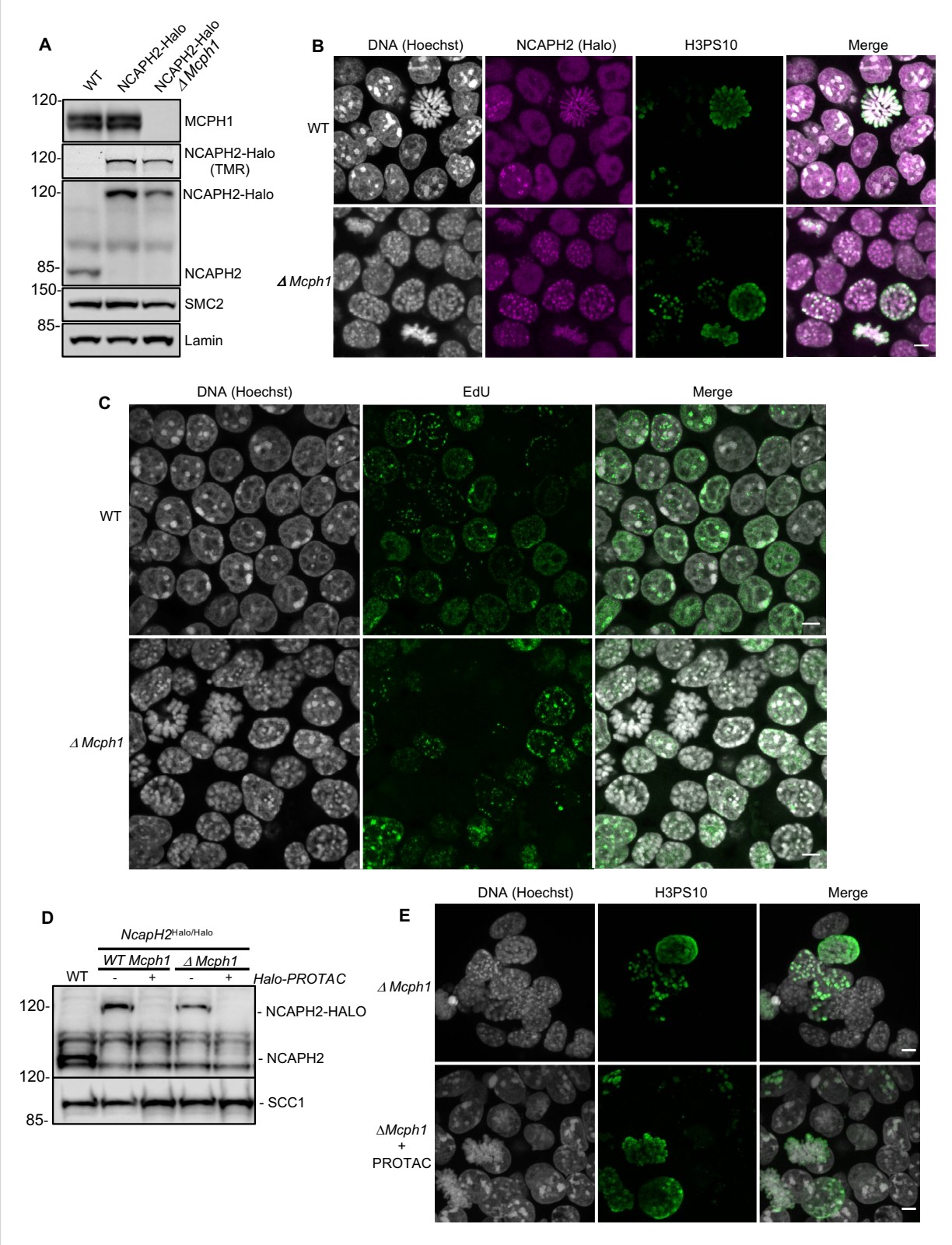

**Figure 1.** *The deletion of Mcph1 in E14 cells induces condensin II-dependent chromosome condensation in both G1 and G2 phases of the cell cycle.* (**A**) In gel TMR-Halo detection and western blot analysis of E14 cells wild type, *Ncaph2^Halo/Halo* and *Ncaph2^Halo/Halo Mcph1^Δ/Δ*. TMR signal detects NCAPH2-Halo tagged. The anti-NCAPH2 antibody shows that all the NCAPH2 protein expressed is fused to the Halo-tag and that the expression levels are similar to wild type but reduced after *Mcph1* deletion. The anti-SMC2 detection shows similar levels of condensin in the three cell lines. (**B**)

*Figure 1 continued on next page*

*Figure 1 continued*

Immunofluorescence analysis of Histone H3 phosphorylated on serine 10 (green) combined with TMR detection of NCAPH2-Halo (Red) in *Mcph1*^wt/wt^ and *Mcph1*^Δ/Δ^ cells. The DNA organisation was analysed using Hoechst. (**C**) EdU incorporation in *Mcph1*^wt/wt^ or *Mcph1*^Δ/Δ^ cells.(**D**) Western blot analysis of Halo-PROTAC induced NCAPH2-Halo degradation in wild-type, *Ncaph2*^Halo/Halo^ *Mcph1*^wt/wt^ and *Ncaph2*^Halo/Halo^ *Mcph1*^Δ/Δ^ cells using an anti-NCAPH2 antibody. Anti-SCC1 was used as a loading control. (**E**) Immunofluorescence analysis of the chromosome decompaction induced by 16 hr treatment of *Mcph1*^Δ/Δ^ cells with Halo-PROTAC. Immunofluorescence analysis of Histone H3 phosphorylated on serine 10 (green) was used to compare similar cell cycle stages. All the cells showed a diffuse chromatin organisation (150 cells counted). Scale bar, 5 μm.

The online version of this article includes the following figure supplement(s) for figure 1:

**Source data 1.** Raw data uncropped blots corresponding to *Figure 1A*.

**Source data 2.** Raw data uncropped blots corresponding to *Figure 1D*.

**Figure supplement 1.** *The cell cycle parameters are unchanged in Mcph1-deleted cells.*

**Figure supplement 1—source data 1.** Raw data uncropped blots corresponding to *Figure 1A*.

**Figure supplement 2.** *Chromocenters disruption in Mcph1 deleted cells.*

**Figure supplement 2—source data 1.** Raw data of foci quantification.

## The chromosome re-organisation induced by *Mcph1* deletion does not depend on CDK1

To check whether the premature formation of prophase-like chromatids when *Mcph1* deleted cells enter G2 might be caused by the precocious activation of CDK1, which activates condensin II as wild-type cells enter prophase, we compared inhibitory phosphorylation of CDK1's Y15 residue and cyclin B1 localisation in wild type and mutant cells. Deletion of *Mcph1* neither reduced Y15 phosphorylation nor caused cyclin B1 to enter nuclei prematurely (*Figure 3A and B*). To address this issue more directly, namely whether CDK1 activity is required for condensin II's hyperactivity, we asked whether inhibition of the kinase would suppress the chromosomal re-organisation. Treatment of both wild type and *Mcph1* deleted cultures with the CDK1 inhibitor RO-3306 for 7 hr caused most cells to accumulate in G2 with a 4 C DNA content (*Figure 3C*). The DNA organisation of wild-type cells resembled that of normal G2 cells, namely no chromatid-like structures were formed and centromeres were clustered in chromocenters (200 cells counted, *Figure 3D*). In contrast, the chromosomal DNAs of all mutant cells were organised into prophase-like chromatids with individualised centromeres (150 cells counted, *Figure 3D*). We conclude that the hyperactivity of condensin II in G2 *Mcph1*-deleted cells is independent of CDK1, suggesting that MCPH1 regulates condensin II directly.

## *Mcph1* deletion induces chromosomal compaction and alters inter-chromosomal interactions

In situ Hi-C libraries were generated for wild-type mouse E14 and *Mcph1* deleted cells. In wild-type cells, chromosomal p-termini have enhanced contact frequencies with one another, as do q-termini. In contrast, p-termini are less likely to contact q-termini (*Figure 4A*). This is consistent with the presence of chromocenters within which p- termini co-localise with each other and likewise q-termini with each other. Because mouse chromosomes are telocentric, with centromeres located at their p-termini, the HiC maps confirm that centromeres co-localise with one another as do telomeres.

Strikingly, the enhanced spatial proximity between chromosomal p-termini (resp., q-termini) is lost in the absence of MCPH1 (*Figure 4B and C*), consistent with the disappearance of chromocenters as observed by microscopy. Deletion of *Mcph1* also results in an enhancement in the frequency of long-range, intra-chromosomal contacts (*Figure 4D, E and F*) and enhances the frequency of inter-compartment (A to B) contacts as compared to intra-compartment contacts (A to A and B to B) (*Figure 4—figure supplement 1A* and B). This finding is consistent with the compaction of individual chromosomes upon loss of MCPH1.

The HiC maps did not reveal loci moving from one compartment to the other nor any major changes in loops or contact domains (*Figure 4—figure supplement 1C*).

## Recombinant MCPH1 forms a stable complex with Condensin II

Our results suggest that MCPH1 directly represses condensin II activity, possibly via a direct interaction. Previous in vitro work suggested that MCPH1 binds condensin II via two interfaces. One interface

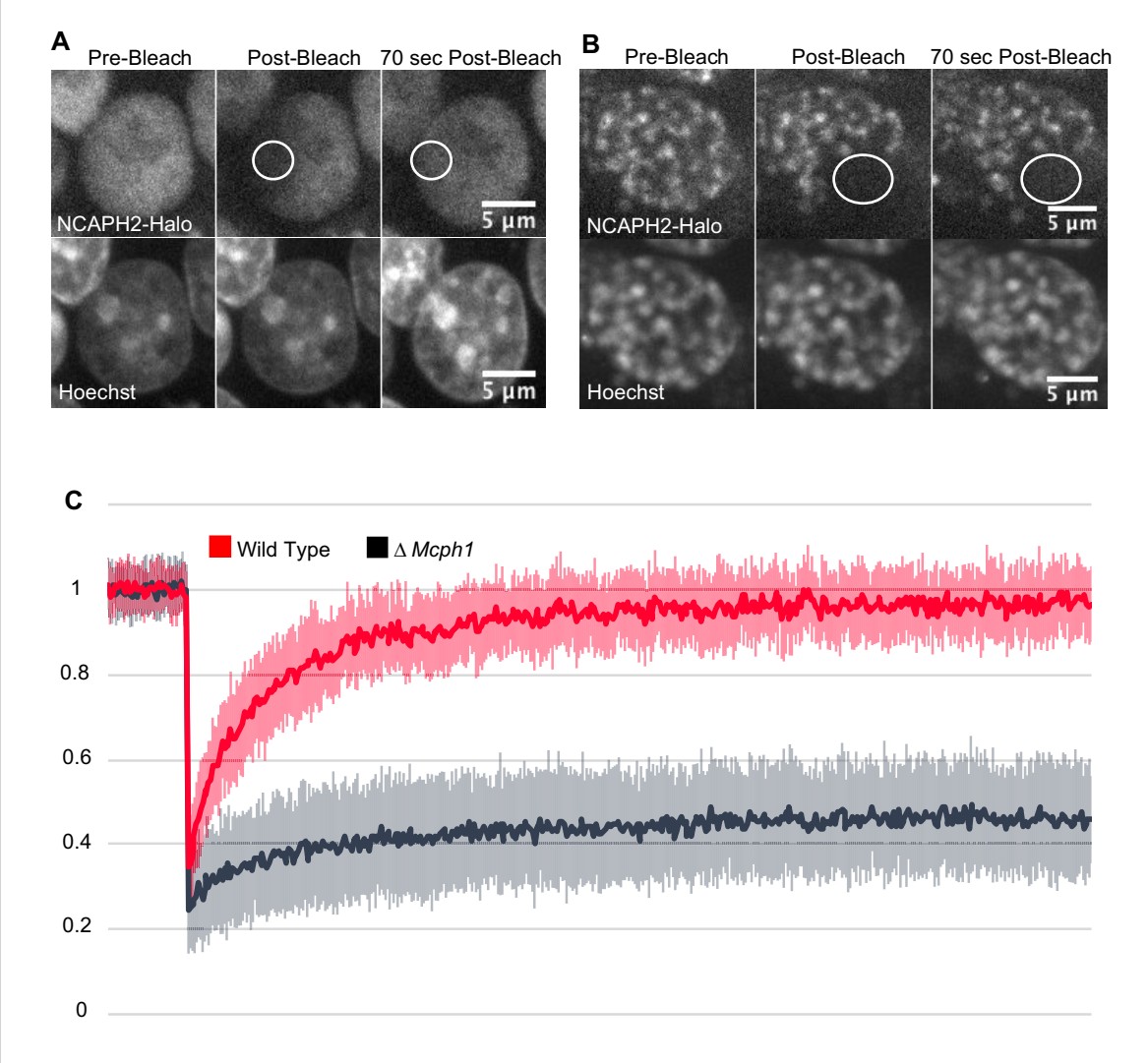

**Figure 2.** *Mcph1 deletion induces the stable binding of condensin II to the condensed chromosomes in interphase.* FRAP analysis of NCAPH2-Halo turn-over on chromatin in *Mcph1^{wt/wt}* (**A**) and *Mcph1^{Δ/Δ}* cells (**B**). Top row: NCAPH2-Halo signal pre-bleach, post bleach and after 70 s recovery. The region bleached correspond to the white circle. Scale bar, 5 µm. (**C**) Quantification of the fluorescence recovery after photobleaching over a 10 min post-bleach period. (Average of three experiments, total number of cells analysed: WT 28 cells, *Mcph1^{Δ/Δ}* 30 cells, standard deviation is represented for every time point).

The online version of this article includes the following figure supplement(s) for figure 2:

**Source data 1.** Microsoft excel of FRAP measurements.

between the N-terminal 195 residues of MCPH1 and NCAPD3 and a second binding site between a highly conserved central domain of MCPH1 (381–435) and NCAPG2 (*Figure 5A and B Yamashita et al., 2011*). To confirm a direct interaction between MCPH1 and condensin II, full-length human MCPH1 and condensin II were expressed in insect cells and separately purified. While full-length MCPH1 was largely insoluble, we purified sufficient strep-tagged full-length MCPH1 to confirm that it could pull-down pentameric condensin II complex (*Figure 5C*). To localise the binding site, we expressed and purified N-terminal His-MBP-tagged truncations of MCPH1 in *E. coli*: MBP-MCPH1$_{1-435}$, MBP-MCPH1$_{1-195}$, MBP-MCPH1$_{196-435}$ and MBP-MCPH1$_{348-469}$ (*Figure 5A*). We found that strep-tagged condensin II could only pull-down MCPH1 constructs that included the central domain (*Figure 5D*). MCPH1 binding was specific to condensin II, as condensin I-strep was unable to pull down any MCPH1 constructs (*Figure 5—figure supplement 1A*).

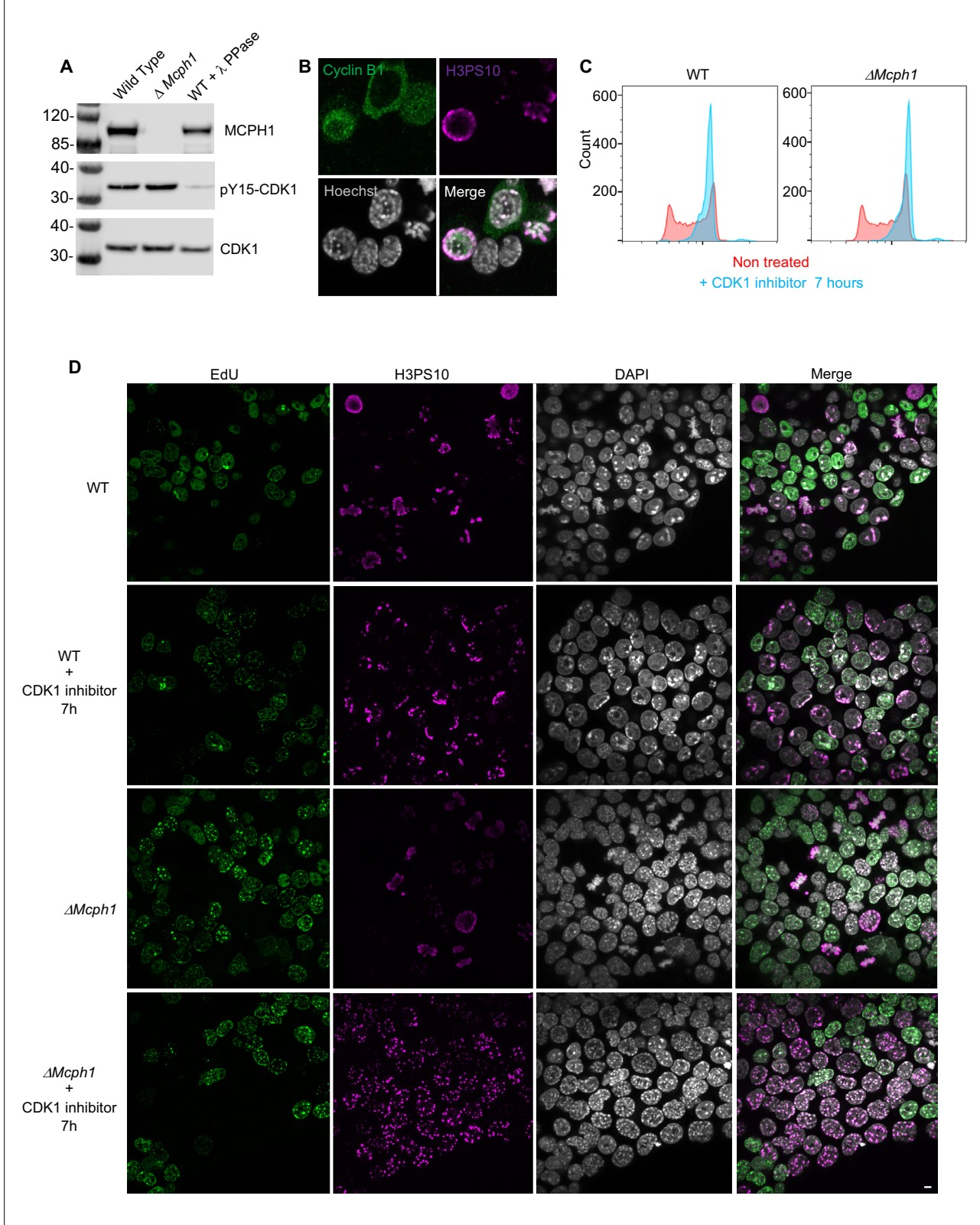

**Figure 3.** *CDK1 activity is not required for the condensation phenotype induced by Mcph1 deletion.* (**A**) Western blot analysis of CDK1 phosphorylation on tyrosine 15 in wild-type cells compared to *Mcph1*-deleted cells. Wild-type protein extracts were treated by λ phosphatase as a control of antibody specificity. An anti-CDK1 protein was used as a loading control. (**B**) Immunolocalisation of Cyclin B in *Mcph1*-deleted cells. (**C**) CDK1 activity was inhibited by incubating wild-type or *Mcph1* deleted cells to RO-3306 for 7 hr. The cell cycle profile was analysed by FACS for both wild-type and *Mcph1*-

*Figure 3 continued on next page*

Figure 3 continued

deleted cells without treatment or after 7 hr incubation with 9 µM R0-3306. (**D**) All the G2 cells in *Mcph1*-deleted cells present the condensed phenotype after CDK1 inhibition (150 cells counted). No condensation was observed in wild-type G2 cells (200 cells counted). Scale bar, 5 µm.

The online version of this article includes the following figure supplement(s) for figure 3:

**Source data 1.** Raw data uncropped blots corresponding to *Figure 3A*.

We then tested whether tetrameric condensin II, lacking either NCAPD3 or NCAPG2, could bind MCPH1$_{1-435}$ using pull-down assays. To exclude the MBP tag interfering with the interaction at the N-terminus of MCPH1, we moved the tag to the C-terminus. We found that removing NCAPG2 greatly reduced MCPH1$_{1-435}$ MBP pull-down, while removing NCAPD3 had no effect (*Figure 5E*), suggesting that the binding was mediated by the central domain of MCPH1 to NCAPG2. Further analysis with analytical size exclusion chromatography demonstrated that MBP-MCPH1$_{196-435}$ and condensin II co-eluted in one peak, separated from the void volume, suggesting they form a stable, soluble complex (*Figure 5—figure supplement 1B*).

## MCPH1 binds Condensin II via a short linear motif

To address which part of MCPH1's central domain is necessary for binding condensin II, we analysed the sequence of MCPH1 using the ConSurf server (*Ashkenazy et al., 2016*; *Berezin et al., 2004*) to identify conserved sequences. Between residues 381–435 of human MCPH1, the 410–424 interval stands out as a highly conserved patch within a region that is otherwise poorly conserved (*Figure 5B* and *Figure 5—figure supplement 1C*). Despite its conservation, the sequences are predicted to be disordered, suggesting that it could be a short linear motif (SLiM) that binds to condensin II. To test this, we performed fluorescence polarisation binding assays with a 5-FAM-labelled MCPH1 peptide spanning residues 407–424 and found that it bound to condensin II, with a fit Kd of 0.64 ± 0.12 µM (mean ± SEM) (*Figure 5F*). As expected, no binding was detected to a tetrameric version of condensin II lacking NCAPG2.

SLiMs are frequently regulated by post-translation modification (*Van Roey et al., 2014*) and proteomic analysis of mitotic cells previously found that MCPH1 can be phosphorylated within the central motif at S417 (*Oppermann et al., 2012*), with S417/P418 forming a potential CDK consensus site (*Errico et al., 2010*). We therefore used a fluorescence polarisation competition assay to test the effect of phosphorylating S417. In these assays, the concentration of condensin II and 5-FAM-MCPH1$_{407-424}$ is fixed and unlabelled peptides of either wild-type or S417 phosphorylated MCPH1$_{407-424}$ are added at increasing concentrations. While the wild-type MCPH1$_{407-424}$ readily competed with 5-FAM-MCPH1$_{407-424}$, resulting in a fit competition K$_D$ of 5.3 ± 1.0 µM, phosphorylation at S417 reduced the affinity ~10 fold to 53 ± 8 µM (*Figure 5G*). This suggests that CDK1 phosphorylation of MCPH1 may reduce its interaction with condensin II, an effect that might have an important role in initiating chromosome condensation during prophase.

## MCPH1 central domain is essential for its interaction with Condensin II and its regulation in vivo

To address whether this central motif is necessary for binding and regulating condensin II in vivo, we created an E14 cell line in which both copies of the *Ncaph2* gene is tagged at its C-terminus with GFP. Western blotting revealed MCPH1 in immunoprecipitates generated using antibodies against GFP, and only in GFP-tagged cells expressing wild-type MCPH1 (*Figure 6A*). Because the MCPH1-specific antibody was raised against the central domain, we used a cell line in which both copies of *Mcph1* and *Ncaph2* were tagged with GFP and Halo respectively to test the role of the central domain motif and then created a variant (*Mcph1$^{ΔCenGFP/ΔCenGFP}$*) lacking 15 residues containing the motif (S$_{400}$SYE-DYFSPDNLKER$_{414}$). Western blotting confirmed that *Mcph1$^{GFP/GFP}$* cells and *Mcph1$^{ΔCenGFP/ΔCenGFP}$* were expressed at similar levels (*Figure 6C*). The slightly increased mobility of *Mcph1$^{ΔCenGFP/ΔCenGFP}$* and its failure to be detected by the MCPH1-specific antibody confirmed deletion of the central domain motif (*Figure 6B*). Because TMR-labelled NCAPH2-Halo was detected in GFP immunoprecipitates from *Mcph1$^{GFP/GFP}$* but not *Mcph1$^{ΔCenGFP/ΔCenGFP}$* cells (*Figure 6C*), we conclude that MCPH1's central domain is essential for its stable interaction with condensin II in vivo.

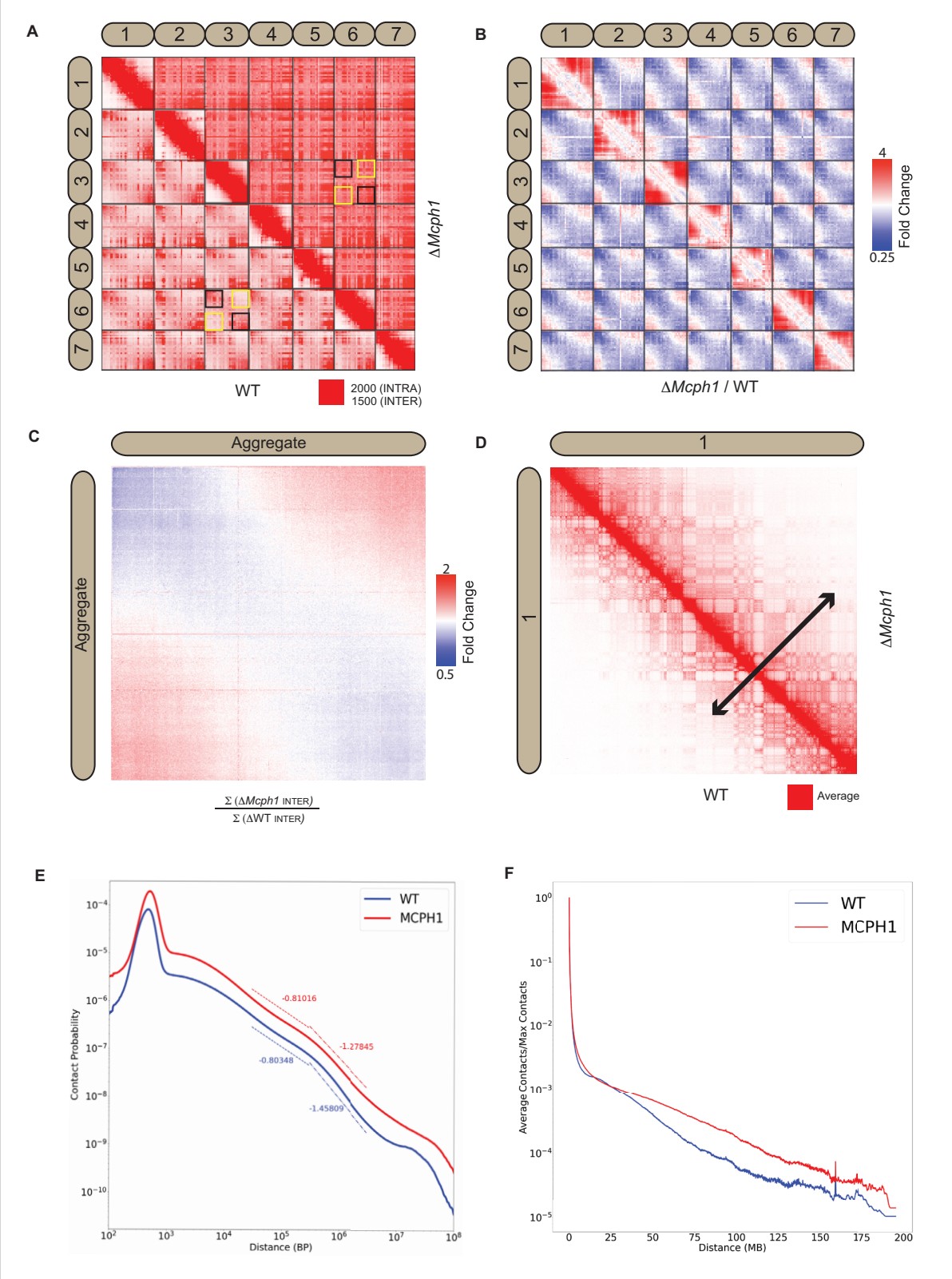

**Figure 4.** *Mcph1 deletion causes chromosome compaction and loss of chromocenters.* (**A**) Representative subset of interactions between chromosomes 1–7 for wild-type and *Mcph1* deletion maps (wild-type below diagonal) shows loss of chromocenters in the *Mcph1* deletion maps. The intensity of color in the Hi-C maps show the frequency of contacts between pairs of loci (row and column), with the upper bound cutoff for interaction frequency in red and zero interaction frequency in white (different upper bound cutoffs for intrachromosomal vs interchromosomal interactions are indicated in

*Figure 4 continued on next page*

*Figure 4 continued*

the legend). In the wild-type map, the p-termini of different chromosomes show increased contact frequency with one another, as do the q-termini of different chromosomes with each other; examples of such interactions are highlighted by the black squares. The wild-type map also shows that the p-termini of one chromosome show less contact frequency with the q-termini of other chromosomes; examples of such interactions are highlighted by the yellow squares. Such differential interactions in the wild-type map are contrasted with the same regions highlighted by the black and yellow squares in the *Mcph1* deletion maps, where no significant difference of interaction is seen. (**B**) Log fold change for *Mcph1* deletion over wild-type for chromosomes 1–7. Red indicates genomic loci pairs that enriched contact in the *Mcph1* deletion maps relative to wild-type, and blue indicates genomic loci pairs that have depleted contacts in the *Mcph1* deletion maps relative to wild-type. (**C**) Log fold enrichment of the *Mcph1* deletion map over wild-type map for the aggregated inter-chromosomal matrix. (**D**) Balanced KR-normalised Hi-C Maps for wild-type and *Mcph1* deletion maps for the intrachromosomal region of chromosome 1 (wild-type below diagonal). Increased interactions between distant loci in the intrachromosomal *Mcph1* deletion maps is seen. Color scale threshold is at the average value of each respective Hi-C map. Black arrows indicate the genomic distance at which loci show a greater than average level of contact frequency for the respective maps. (**E**) Intrachromosomal contact probability for all chromosomes shows increased long-range interactions and diminished contact drop-off for *Mcph1* deletion. (**F**) Average intrachromosomal contact frequency for all chromosomes shows increased long-range interactions with *Mcph1* deletion.

The online version of this article includes the following figure supplement(s) for figure 4:

**Figure supplement 1.** *Mcph1 deletion decreases intra-compartment strength and does not affect looping.*

Immunofluorescence revealed that MCPH1 and MCPH1$^{\Delta Cen}$-GFP proteins show the same localisation within cells. They are both exclusively nuclear and enriched in small clusters that colocalise with sites of DNA damage marked by γH2aX DNA (*Figure 6D*), as previously reported (*Rai et al., 2006*). The significance of this association is unclear as *Mcph1* deletion has no effect on the level of γH2aX (*Figure 1—figure supplement 1D*). Crucially, the chromosomes of *Mcph1$^{\Delta CenGFP/\Delta CenGFP}$* but not *Mcph1$^{GFP/GFP}$* G1 and G2 cells adopted the prophase-like appearance characteristic of *Mcph1* deleted cells (*Figure 6E*). Therefore, we conclude that MCPH1's central domain SLiM is essential for inhibiting condensin II during interphase and inhibiting premature condensin-mediated chromatin condensation.

## MCPH1 does not alter Condensin II ATPase activity or DNA binding in vitro

To address whether MCPH1 affects condensin II's activity in vitro, purified the condensin II- MCPH1$_{1-435}$ complex using size exclusion chromatography and measured its ATPase activity. The condensin II-MCPH1$_{1-435}$ complex possessed a similar activity to that of condensin II alone but its stimulation by DNA was modestly lower (*Figure 7A*). To ensure that the ATPase activity measured in these assays was genuinely due to condensin II, we also purified a condensin II-MCPH1$_{1-435}$ complex deficient in ATP binding (Q-loop mutation, SMC2 Q147L, SMC4 Q229L). As expected, this mutation effectively eliminated ATPase hydrolysis (*Figure 7A*).

Previous work has suggested that full-length MCPH1 binds to DNA and chromatin (*Chang et al., 2020*; *Yamashita et al., 2011*) so we tested whether MBP-MCPH1$_{1-435}$ and MBP-MCPH1$_{196-435}$ are able to bind to a 50 bp sequence of dsDNA using electrophoretic mobility shift assay (EMSA). Both MCPH1$_{1-435}$-MBP and MCPH1$_{196-435}$-MBP were able to induce a shift, however higher concentrations of MCPH1$_{196-435}$ MBP were required for a complete shift in the free DNA band, suggesting MCPH1$_{1-195}$ MBP could have a role in DNA binding (*Figure 7B*). We then examined if MCPH1 affected condensin II DNA binding, by performing condensin II EMSAs in the presence or absence of MCPH1$_{1-435}$-MBP. The distinct MCPH1$_{1-435}$-MBP shifted band disappeared with increasing concentrations of condensin II, and there was an upward shift in the condensin II bands in the presence of MCPH1$_{1-435}$-MBP relative to the MBP condensin II control, suggesting MCPH1 was binding with condensin II (*Figure 7C*). We then performed condensin II EMSAs in the presence or absence of 1 µM 5-FAM-MCPH1$_{407-424}$ peptide. 5-FAM signal was present with condensin II shifted DNA band demonstrating that the MCPH1 peptide was sufficient to mediate comigration of MCPH1 with condensin II (*Figure 7D*). Additionally, the presence of the 5-FAM-MCPH1$_{407-424}$ peptide did not affect condensin II DNA binding. Collectively, this indicates that in vitro, using purified proteins, condensin II can bind MCPH1 and DNA simultaneously. Finally, these results suggest that the inhibitory effect of MCPH1 on condensin II loading observed in vivo involves a feature absent from the in vitro DNA binding assay.

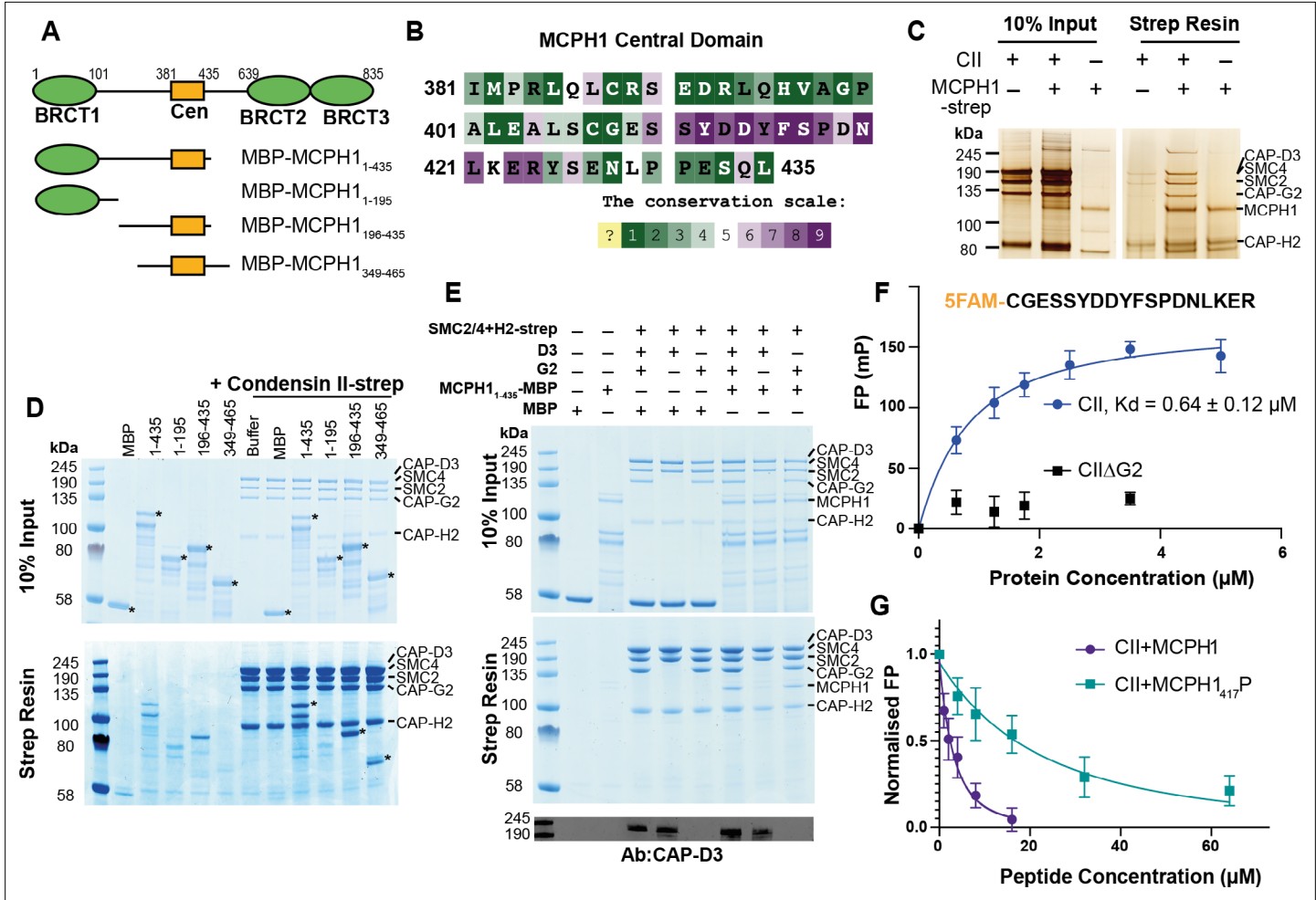

**Figure 5.** *Human condensin II interaction with MCPH1.* (**A**) Domain structure of *MCPH1* and *MBP* fusion constructs that were expressed in *E. coli* and used in binding assays. BRCT domains are indicated in green and the central domain in yellow. (**B**) Conservation analysis of the MCPH1 central domain with the ConSurf server. (**C**) Strep tag pull-down indicating full-length MCPH1 binds condensin II. Full-length MCPH1 and condensin II were expressed in insect cells and separately purified, before being mixed on strep-tactin sepharose. Samples of input and resin after run on SDS page and visualised with silver stain. (**D**) Strep-tag pull-down assay indicating strep tagged condensin II pulls down MBP-MCPH1 constructs that contain the central domain, but not MBP-MCPH1$_{1-195}$ or MBP alone. SDS page gel visualised with Coomassie stain, (*) indicates the running position of the MBP/MCPH1 construct used. (**E**) Strep pull-down assay showing strep-tagged pentameric condensin II or tetrameric condensin II lacking NCAPD3 can pull down MBP-MCPH1$_{1-435}$, while tetrameric condensin lacking NCAPG2 does not pull down MCPH1. The lower panel shows a western blot performed using strep-resin samples, blotted using an anti-NCAPD3 antibody. (**F**) Fluorescence polarisation binding assay using 5-FAM-labelled MCPH1$_{407-424}$ peptide and increasing concentration of either pentameric condensin or tetrameric condensin II lacking MCPH1 binding subunit NCAPG2 (CIIΔG2). (**G**) Peptide competition assay using a fixed concentration of 5-FAM labelled MCPH1$_{407-424}$ and condensin II with an increasing amount of MCPH1$_{407-424}$ wild-type or phosphorylated at serine 417. All error bars indicate standard deviation from three replicates.

The online version of this article includes the following figure supplement(s) for figure 5:

**Source data 1.** Microsoft excel data corresponding to *Figure 5F*.

**Source data 2.** Microsoft excel data corresponding to *Figure 5G*.

**Source data 3.** Raw data uncropped gels corresponding to *Figure 5C, D and E*.

**Figure supplement 1.** *Condensin I does not interact with MCPH1.*

**Figure supplement 1—source data 1.** Raw data uncropped gels corresponding to *Figure 5—figure supplement 1B, C*.

**Figure supplement 1—source data 2.** Microsoft excel data corresponding to *Figure 5—figure supplement 1B*.

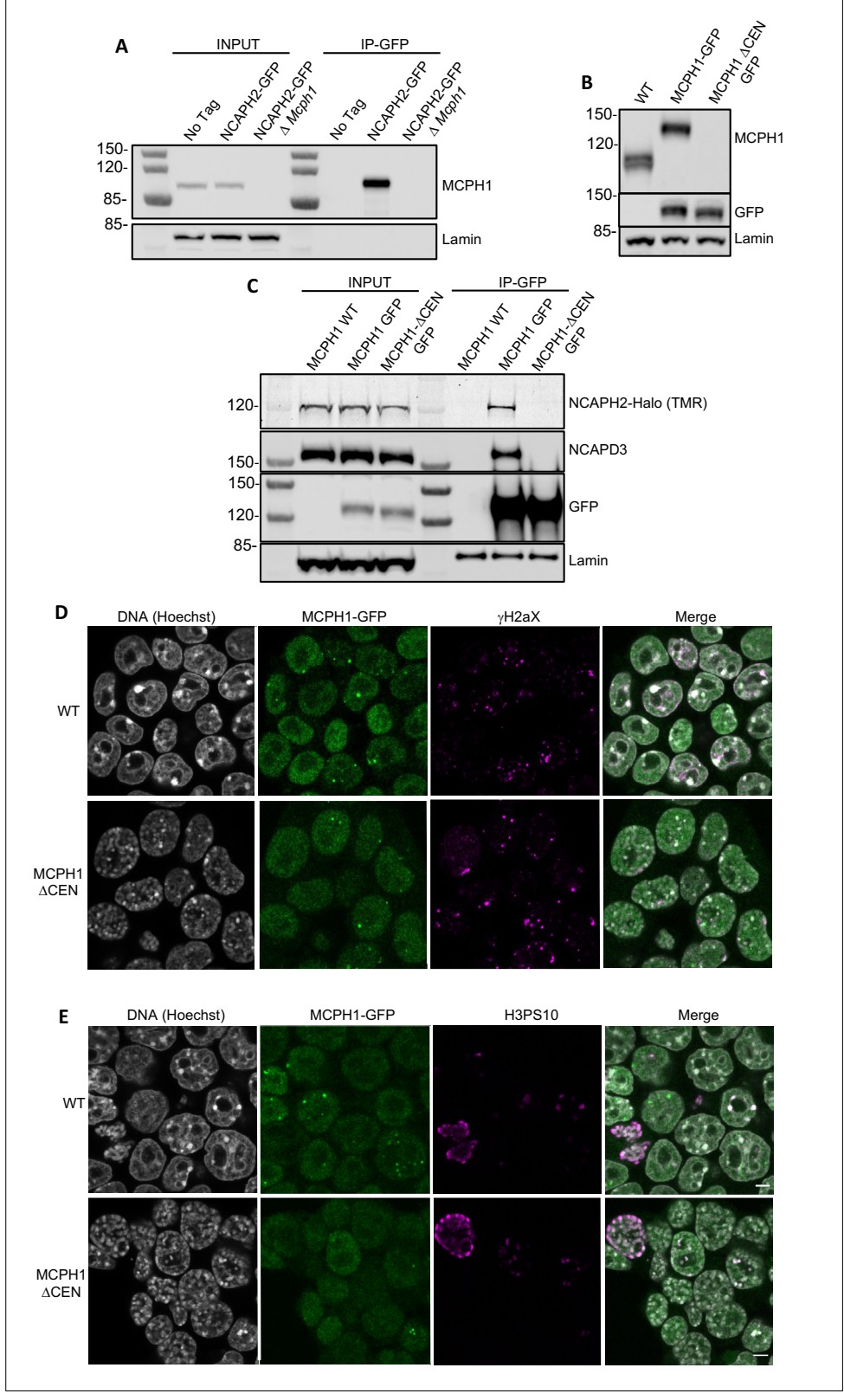

**Figure 6.** *MCPH1 interaction with condensin II is essential to prevent interphasic chromosome condensation.* (**A**) Co-immunoprecipitation of MCPH1 with NCAPH2-GFP. Nuclear extracts were prepared from wild type, *Ncaph2^{GFP/GFP}* and *Ncaph2^{GFP/GFP} Mcph1^{Δ/Δ}* cells. Immunoprecipitation was performed using GFP-trap agarose beads and analysed by western blot using an anti-MCPH1 antibody (IP-GFP). 5 % of the lysate used for IP was loaded as

*Figure 6 continued on next page*

*Figure 6 continued*

INPUT control. Anti-Lamin B1 antibody was used as a loading control. (**B**) Deletion of the central domain of MCPH1. To address if the central domain of MCPH1 is necessary to mediate the interaction with condensin II, we first introduced a GFP-tag at the C-terminal end of MCPH1 in *Ncaph2^Halo/Halo* cells as the antibody against the protein was raised against the central domain. Then a second targeting was done to delete the central domain. As a result, the western blot represented in panel B using anti-MCPH1 antibody detects the wild-type protein or the GFP-tagged protein, homozygous *Mcph1^GFP/GFP* but does not detect anything after deletion of the central domain. Using anti-GFP antibody reveals that the protein deleted for the central domain is present in the cell at similar levels as the wild-type GFP-tagged protein. A slight decrease in size is observed due to the deletion of the central domain. Anti-Lamin B1 antibody was used as a loading control. (**C**) Co-immunoprecipitation of NCAPH2-Halo with MCPH1-GFP. Nuclear extracts were prepared from *Ncaph2^Halo/Halo*, *Ncaph2^Halo/Halo Mcph1^GFP/GFP* and *Ncaph2^Halo/Halo Mcph1^ΔcenGFP/ΔcenGFP* cells. Immunoprecipitation was performed using GFP-trap agarose beads and analysed by in-gel detection of NCAPH2-Halo using the Halo-ligand TMR or by western blot using an anti-NCAPD3 antibody (IP-GFP). 5 % of the lysate used for IP was loaded as input control. Anti-Lamin B1 antibody was used as a loading control. (**D,E**) Immunofluorescence analysis of the chromatin organisation in *Ncaph2^Halo/Halo Mcph1^GFP/GFP* and *Ncaph2^Halo/Halo Mcph1^ΔcenGFP/ΔcenGFP* cells. MCPH1-GFP is only detected in the cell nucleus enriched in dots colocalising with some γH2AX foci (**D**). The deletion of its central domain induces a similar condensation of interphasic chromosomes as the one observed after the complete loss of function of *Mcph1* (**E**).

The online version of this article includes the following figure supplement(s) for figure 6:

**Source data 1.** Raw data uncropped gels corresponding to *Figure 6A*.

**Source data 2.** Raw data uncropped gels corresponding to *Figure 6B*.

**Source data 3.** Raw data uncropped gels corresponding to *Figure 6C*.

## MCPH1 overexpression inhibits the loading of Condensin II on mouse meiotic chromosomes

Our results show that MCPH1 plays a crucial role in chromosome organisation during interphase by inhibiting condensin II's activity in mitotic cells. We next extended our analysis of MCPH1 to meiotic cells by testing the effect of increased MCPH1 expression during the first meiotic division of mouse oocytes. To image the chromosomes, oocytes were injected with in vitro transcribed mRNA coding for H2B-mCherry to illuminate the chromosomes in magenta (*Figure 8A*). As previously described (*Houlard et al., 2015*), after the germinal vesicle breakdown (GVBD), bivalent chromosomes form a ball (3.3 hr) before congressing to a metaphase plate (7.4 hr). Cleavage of cohesin by separase along chromosome arms then converts each bivalent into a pair of dyads that segregate highly synchronously to opposite poles of the cells during anaphase I (10 hr), which is followed by extrusion of the first polar body (*Figure 8B*, control).

Co-injection with *M. musculus Mcph1* mRNAs (*Figure 8B*,+ MCPH1) had little effect on the chromosome congression until metaphase. However, it caused chromosomes to unravel soon after the onset of anaphase, presumably because chromosomes are not stiff enough to resist to the pulling forces of the spindle, and this was accompanied by a catastrophic failure to disjoin the chromosome arms of dyads to opposite poles (*Figure 8B*,+ *Mcph1* mRNA, 16 h). The lack of chromosome rigidity and the unravelling of the chromatin in response to traction by the spindle are reminiscent of the phenotype caused by depletion of NCAPH2 (*Houlard et al., 2015*).

To address whether MCPH1 overexpression inhibits the association of condensin II with chromosomes, we rescued mouse oocytes deleted for *Ncaph2* (*Ncaph2^Lox/Lox*, *Zp3^TgCre*) by injecting an mRNA coding for NCAPH2-GFP as previously described (*Houlard et al., 2015*). These oocytes were also injected with mRNA encoding MAD2 to arrest them in meiosis I and H2B-mCherry to image and quantify the amount of chromosomal NCAPH2-GFP (*Figure 8C*). This revealed that the injection of *Mcph1* mRNAs greatly reduced association of condensin II with chromosomes during metaphase I (*Figure 8D and E*). To see if MCPH1 can also release condensin II previously associated with chromosomes, we injected *Mcph1* mRNA into meiosis I arrested oocytes. However, this induced a much milder reduction within the first two hours (not shown), suggesting that *Mcph1* prevents the initial loading of condensin II on chromosomes but has little effect on complexes already stably associated with them.

The N-terminal BRCT domain of MCPH1 was previously shown to have an essential role in regulating chromosome condensation both in vitro and in vivo (*Yamashita et al., 2011*). Consistent with this, deletion of the N-terminal 200 amino acids of MCPH1 abolishes its inhibitory effect in mouse

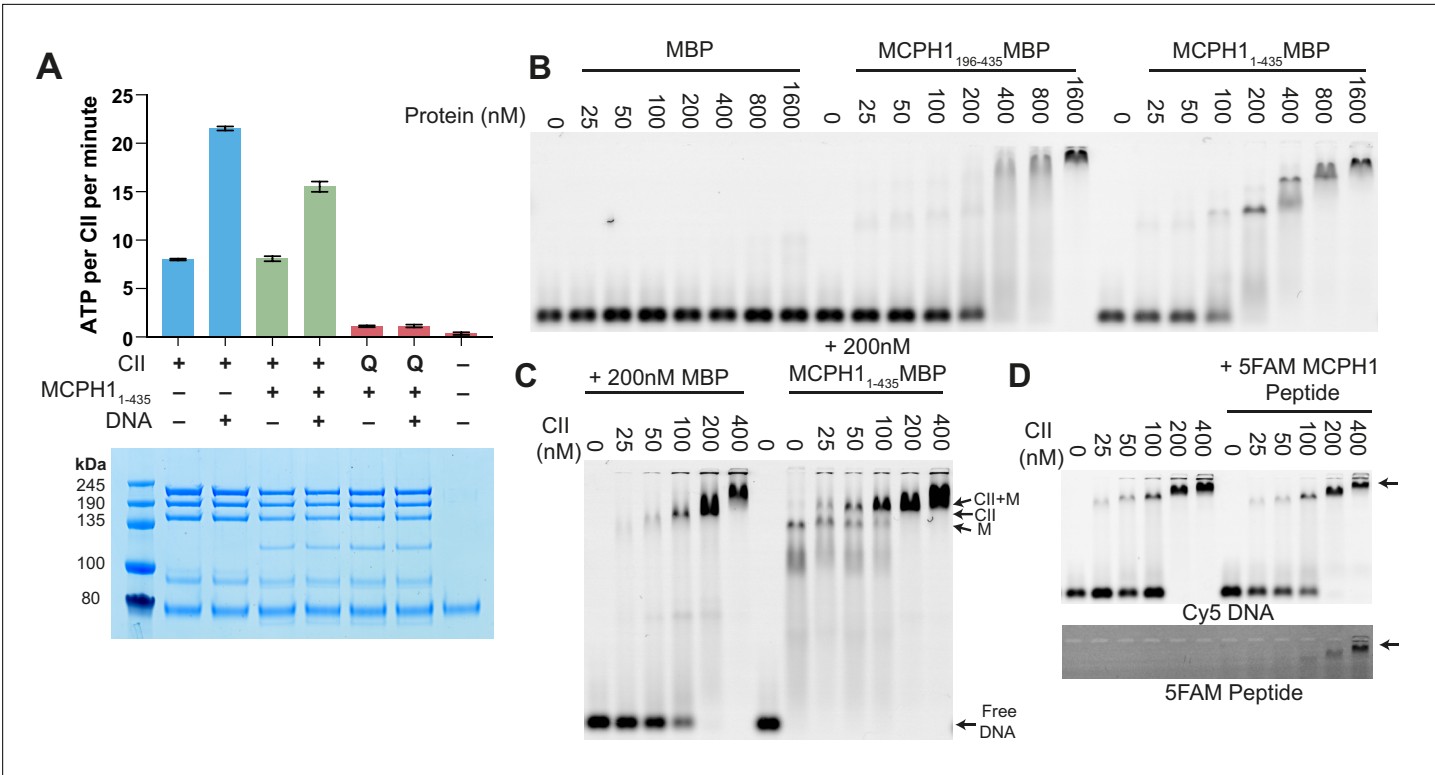

**Figure 7.** *MCPH1 has little effect on condensin II ATP hydrolysis and DNA binding.* (**A**) ATPase rate of condensin II complex in the presence of MCPH1. Q refers to condensin II with an ATPase deficient mutation in the Q-loop. Below is an SDS page gel of the completed reaction. Error bars indicate standard deviation from three repeats. (**B**) EMSA assay of MBP, MCPH1$_{1-435}$-MBP and -MCPH1$_{195-435}$-MBP using 50 bp of Cy5-labelled dsDNA. (**C**) EMSA assay of condensin II in the presence of MBP or MCPH1$_{1-435}$-MBP. (**D**) EMSA assay of condensin II in the presence or absence of 5FAM-MCPH1 peptide. Top image detecting Cy5 and bottom imaged detecting 5FAM.

The online version of this article includes the following figure supplement(s) for figure 7:

**Source data 1.** Microsoft excel data corresponding to the standard curve in *Figure 7A*.

**Source data 2.** Microsoft excel data corresponding to the ATPase curve in *Figure 7A*.

**Source data 3.** raw data uncropped gels corresponding to *Figure 7*.

oocytes (*Figure 8D*). It has also been claimed that the N-terminal domain of MCPH1 can on its own inhibit condensin II by competing for its binding sites on chromosomal DNA (*Yamashita et al., 2011*). However, we were unable to detect any impact of over-expressing only the NTD of MCPH1 (not shown).

## Fusion of SMC2 to NCAPH2 is resistant to MCPH1 inhibitory effect

The effects of MCPH1 on condensin II resemble those of WAPL on cohesin, which is thought to act by dissociating the NTD of its kleisin from the neck of SMC3's ATPase domain (*Beckouët et al., 2016*; *Chan et al., 2012*; *Eichinger et al., 2013*). A key finding in this regard is that the fusion of the C-terminus of SMC3 to the N-terminus of SCC1 causes cohesin to resist WAPL. To address whether fusion of this nature has similar effect on condensin II, we created a cDNA encoding a protein in which the C-terminus of SMC2 and the N-terminus of NCAPH2 are connected by a 57 amino acid linker containing three TEV protease cleavage sites. A GFP tag was introduced at the C-terminal end of NCAPH2 to image the protein (*Figure 9A*). Importantly, mRNAs encoding this fusion fully rescued the meiosis I chromosome segregation defects of oocytes deleted for *Ncaph2*: 17 out of 18 oocytes deleted for *Ncaph2* and injected with the fusion segregated their chromosome without any defect, whereas all the *Ncaph2* deleted oocytes showed chromosome stretching (n = 10). Furthermore, GFP fluorescence associated with the fusion protein was detected along the chromosome axes of bivalent chromosomes, a distribution that is similar if not identical to that of wild-type NCAPH2. Remarkably, co-injection of *Mcph1* mRNAs had no adverse effect on this activity, unlike controls in

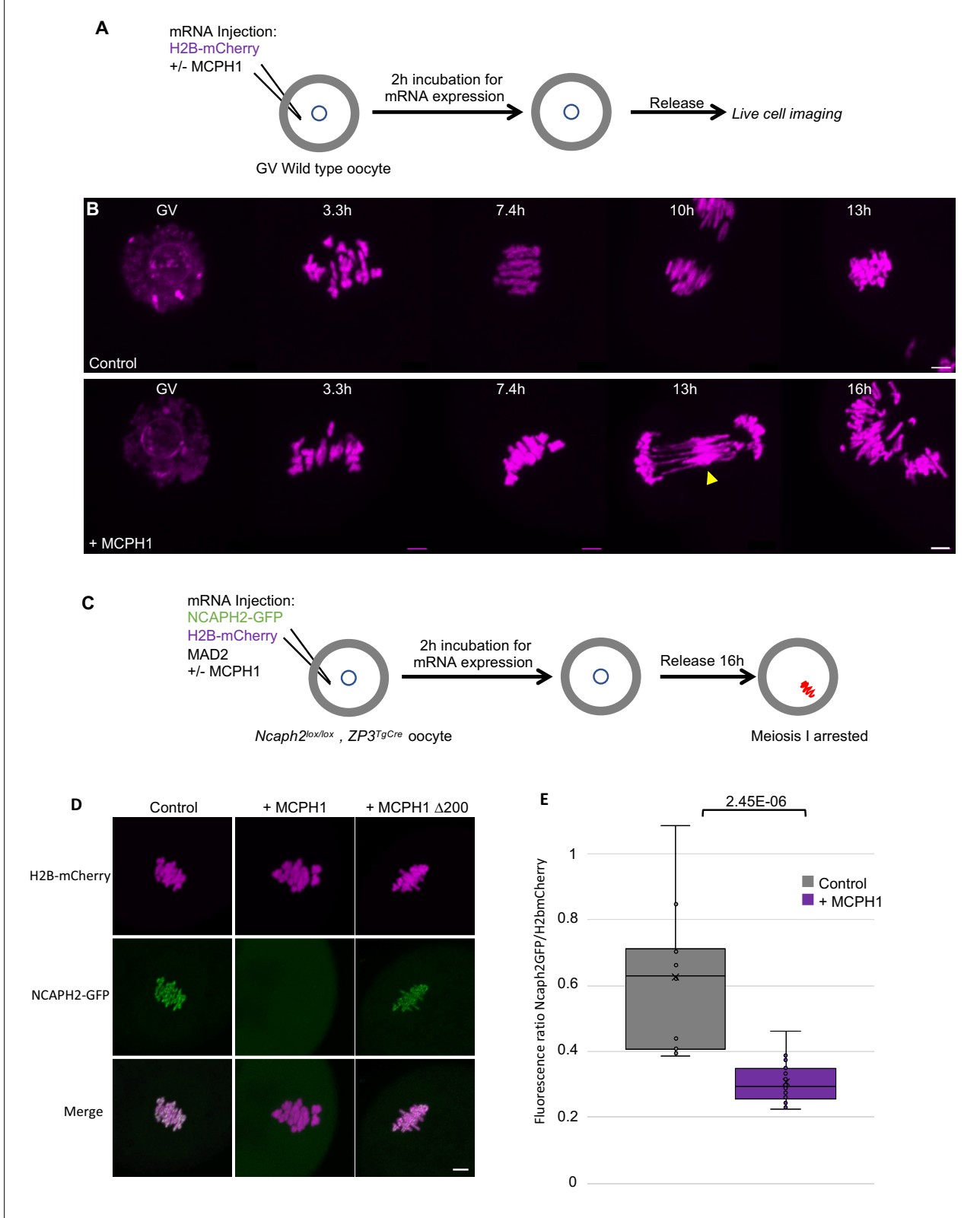

**Figure 8.** *MCPH1 prevents the association of condensin II with chromosomes.* (**A**) Cartoon summarizing the experimental procedure corresponding to panel B. (**B**) Wild-type mouse oocytes were injected at the GV stage with in vitro transcribed mRNA coding for H2B-mCherry alone to mark the chromosomes in magenta (Control) or in combination with MCPH1 (+ MCPH1). Meiosis I progression was followed by live cell confocal imaging. The segregation defects observed in the presence of MCPH1 are indicated by a yellow arrowhead. Maximum intensity z projection images of the main time

*Figure 8 continued on next page*

*Figure 8 continued*

points are shown between 3.3 hr post GVBD onwards (number of oocytes analysed in three independent experiments and showing segregation defects: control:0/12;+ MCPH1: 18/19). (**C**) Cartoon summarizing the experimental procedure corresponding to panel D. (**D**) Oocytes from *Ncaph2^f/f^ Tg(Zp3Cre)* females were injected at the GV stage with mRNA coding for H2B-mCherry, MAD2 and NCAPH2-GFP only (control) or in combination with MCPH1 (+ MCPH1) or MCPH1 deleted of the first N-terminal 200 amino acids (+ MCPH1 Δ200). Oocytes were arrested 16 hr after GVBD in metaphase I owing to MAD2 overexpression, and maximum-intensity z projection images of chromosomes were acquired by live cell confocal imaging (Total number of oocytes analysed in three experiments: control: 37,+ MCPH1: 58,+ MCPH1Δ200: 12). (**E**) Quantification of NCAPH2-GFP signal on the chromosomes. (Number of oocytes analysed in two independent experiments: control: 11;+ MCPH1: 14). In each graph the two tailed T-test p values are indicated is indicated. Scale bar, 5 μm.

The online version of this article includes the following figure supplement(s) for figure 8:

**Source data 1.** Microsoft excel file of fluorescence measurements in *Figure 8E*.

which *Mcph1* mRNAs were co-injected with *Ncaph2-Gfp* mRNAs (*Figure 9B and C*). Likewise, MCPH1 prevented association of NCAPH2-GFP with chromosomes but not that of the SMC2-NCAPH2-GFP fusion (*Figure 9C*).

A caveat to this experiment is that the resistance to MCPH1 of the SMC2-NCAPH2-GFP fusion could be due to the linker sequences associated with the N- and C-termini of NCAPH2 and SMC2 respectively rather than their stable inter-connection per se. If the latter were the case, cleavage of the linker using TEV protease should restore sensitivity to MCPH1. We therefore repeated the rescue experiment but, in this case co-injected mRNA encoding TEV protease (*Figure 9D*). Quantification of the chromosomal GFP fluorescence showed that the fusion's resistance to MCPH1 activity is abolished by the TEV (*Figure 9E and F*), which is consistent with the notion that resistance arises from connecting the interface between SMC2 and NCAPH2.

## *Mcph1* deletion induces the coiling of Cohesin vermicelli

Our results suggest that condensin II's association with chromosomal DNA might be regulated by MCPH1 through a mechanism that resembles that of cohesin by WAPL. By inducing cohesin's dissociation from chromatin, albeit only rarely approximately every 15 min., WAPL merely moderates the processivity of loop extrusion mediated by cohesin. MCPH1 has a more drastic effect, preventing most condensin II complexes from ever associating stably with chromatin. The entire architecture of interphase chromatin therefore depends on these two key regulatory factors. Although condensin II is presumed like cohesin to act as a DNA loop extruder, the deregulation of cohesin and condensin II induces different chromosomal morphologies. *Wapl* deletion enables cohesin to form thread-like structures and to accumulate along their longitudinal axes, creating so called vermicelli. In contrast, *Mcph1* deletion enables condensin II to produce soft spherical or 'gumball' chromosomes and to associate stably throughout chromosomal DNA, albeit at high levels at centromeres. Strangely, condensin II does not form or accumulate along the sort of axes observed when wild type cells enter mitosis. Thus, the activities of condensin II and cohesin unleashed by *Mcph1* and *Wapl* deletion produce DNA loops with very different arrangements, respectively. We therefore set out to address two questions. First, is this difference intrinsic to differences in the behaviour of cohesin and condensin II or merely due to differences in the type of cell used, namely mouse fibroblasts and ES cells? Assuming that it is in fact, the former, what happens when both factors are deregulated simultaneously? To this end, we altered the *Mcph1* and *Wapl* genes in E14 cells in which SCC1 is tagged with Halo and NCAPH2 with GFP. Because *Wapl* deletion is lethal, we generated a tamoxifen-inducible deletion allele, which enabled us to compare chromosomal DNA morphology as well as localisation of SCC1-Halo and NCAPH2-GFP in four different conditions: Wild type, *ΔWapl*, *ΔMcph1* and the double mutation *ΔWapl, ΔMcph1* (*Figure 10—figure supplement 1*).

In wild-type cells, SCC1-Halo accumulates throughout nuclei and their genomes during interphase and apart from centromeres, is largely removed from chromosomes through the action of WAPL in M phase. Condensin II's distribution resembles that of cohesin throughout most of interphase (*Figure 2* and *Gerlich et al., 2006*). Condensin II accumulates around centromeres during G2 and along the chromatid axes created through its activity during prophase (*Figure 10*, WT).

As previously reported for fibroblasts, *Wapl* deletion in E14 cells causes cohesin to create chromatid-like structures, especially during G2, and to accumulate along their longitudinal axes (*Tedeschi et al., 2013*). Because the majority of SCC1 remains on chromosomes during mitosis, most is cleaved by

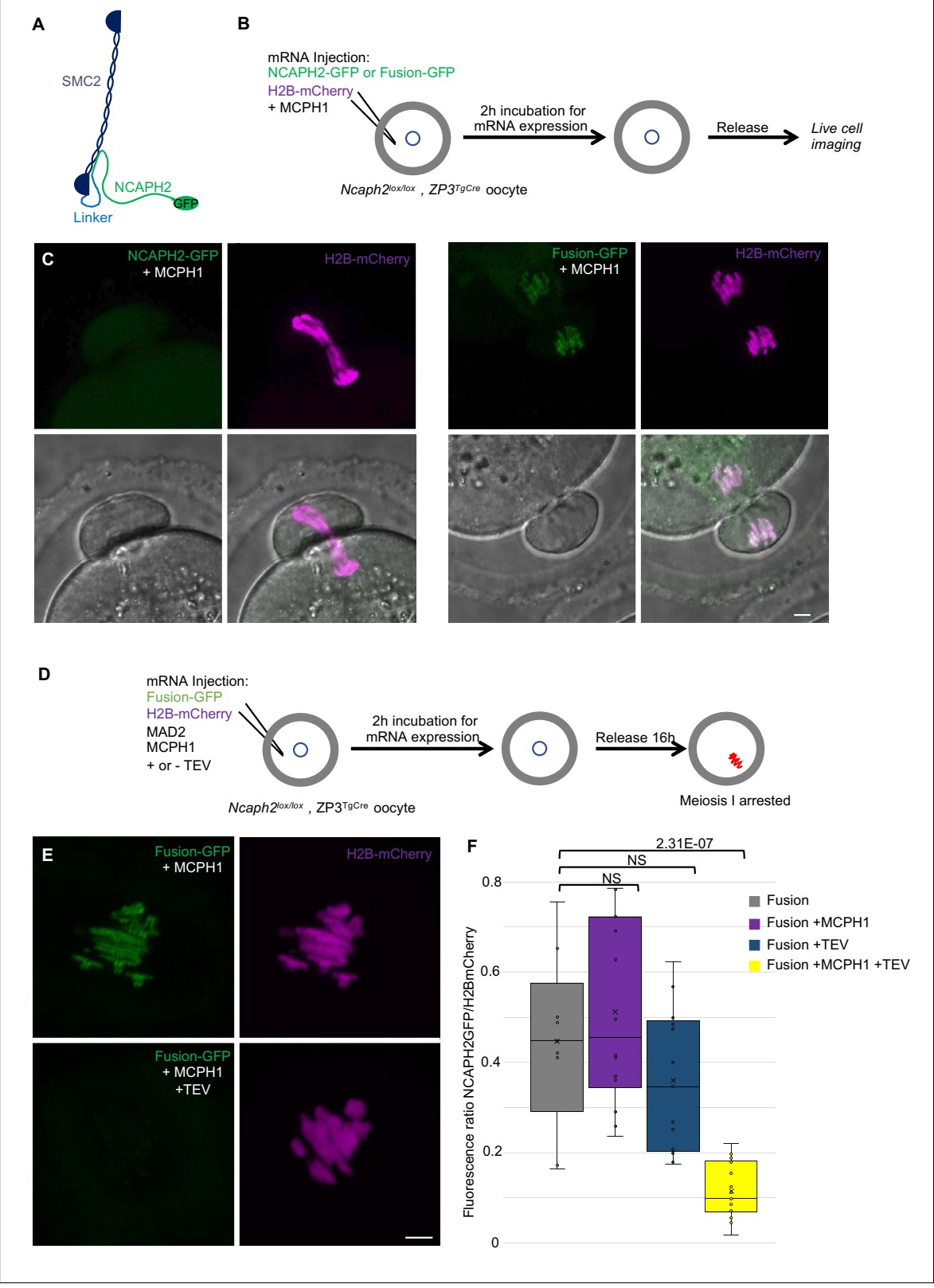

**Figure 9.** *The closure of the SMC2-NCAPH2 interface prevents MCPH1 inhibitory effect.* (**A**) Schematic representation of the protein fusion between SMC2 C-terminus and the N-terminus of NCAPH2 using a linker comprising three TEV protease cleavage sites. (**B**) Cartoon summarizing the experimental procedure corresponding to panel C. (**C**) Oocytes from *Ncaph2^{f/f} Tg(Zp3Cre)* females were injected at the GV stage with mRNA coding for H2B-mCherry and MCPH1 in combination with NCAPH2-GFP (NCAPH2-GFP+ MCPH1) or with the fusion (Fusion-GFP+ MCPH1). Meiosis I progression

*Figure 9 continued on next page*

*Figure 9 continued*

was followed by live cell confocal imaging. Maximum intensity z-projections images of the time points corresponding to anaphase I when segregation defects are observed (total number of oocytes showing chromosome segregation defects in three experiments: NCAPH2-GFP+ MCPH1: 17/19, Fusion-GFP+ MCPH1: 0/18). (D) Cartoon summarizing the experimental procedure corresponding to panel E. (E) Oocytes from *Ncaph2*$^{f/f}$ *Tg(Zp3Cre)* females were injected at the GV stage with mRNA coding for H2B-mCherry, MAD2 and Fusion-GFP only or in combination with TEV protease. Oocytes were arrested 16 hr after GVBD in metaphase I owing to MAD2 over-expression and maximum-intensity z-projection images of chromosomes were acquired by live cell confocal imaging. (F) Quantification of Fusion-GFP signal on the chromosomes (Total number of oocytes analysed in three experiments: Fusion: 9; Fusion+ MCPH1: 14, Fusion+ TEV: 13, Fusion+ TEV + MCPH1: 18). In each graph the two tailed T-test p values are indicated is indicated. Scale bar, 5 μm.

The online version of this article includes the following figure supplement(s) for figure 9:

**Source data 1.** Microsoft excel file of fluorescence measurements in *Figure 9F*.

separase during anaphase and daughter cells inherit considerably less cohesin than normal. As a consequence, axial cohesin vermicelli are rarely if ever observed during G1, especially as this cell cycle phase is very short in ES cells. Pronounced vermicelli are only observed in G2. As expected, *Wapl* deletion neither alters condensin II's distribution during interphase nor hinders its accumulation along chromatid axes during mitosis (*Figure 10—figure supplement 2*). Despite cohesin's persistence on mitotic chromosomes and the participation of a large fraction in sister chromatid cohesion, condensin II still manages to localise to and help create the axes of individual chromatids, between which run inter-chromatid axes coated in cohesin (*Figure 10A and B*, Δ*Wapl*). There are presumably two pools of chromosomal cohesin in post-replicative *Wapl*-deleted cells, one involved in cohesion and a second engaged in the loop extrusion responsible for the formation of vermicelli. The former clearly persists and accumulates along the inter-chromatid axis when loop extrusion mediated by condensin I and II individualise chromatids, but the fate of the latter is unclear.

Deletion of *Mcph1* had little effect on cohesin's distribution. Despite the formation of gumball chromosomes during G1 and G2, cohesin remains uniformly associated with chromatin and does not form vermicelli. As in wild-type cells, most dissociates from chromosome arms when cells enter mitosis and little can be detected along the inter-chromatid axes connecting the two condensin II axes of individualised chromatids (*Figure 10*, Δ*Mcph1*).

Deletion of both *Wapl* and *Mcph1* had a dramatic effect. The cohesin vermicelli caused by the lack of WAPL in G2 cells adopt a coiled configuration upon the simultaneous deletion of *Mcph1*. This coiling increases during prophase. By the time cells reach metaphase, the coiling leads to the formation of chromosomes that have the shape of a spring (or solenoid), a configuration that is visible with SCC1-Halo, NCAPH2-GFP and DNA staining (DAPI) (*Figure 10*, Δ*Wapl,*Δ*Mcph1*). Moreover, the two distinct axes of condensin II associated with each chromatid remain intermingled in the double mutant.

To analyse the axial organisation of these chromosomes in greater detail, we used super-resolution three-dimensional structured illumination microscopy (3D-SIM) to compare the distribution of SCC1-Halo in *Wapl* deleted and double mutant cells. Unfortunately, fluorescence due to NCAPH2-GFP was insufficient to reveal reliable images using this technique. Analysis of G2 cells revealed that a modest coiling of cohesin axes surrounded by DNA loops in *Wapl* single mutants is greatly accentuated in double-mutant cells, with a pronounced increase in the radii of coils (*Figure 11* and *Figure 11—video 1*). In metaphase cells, the cohesin axes that seem to have a spring-like appearance in confocal microscopy are revealed to have a much more complicated organisation, being composed of twisted segments that regularly change handedness. It is noticeable that the formation of chromosomes with this morphology is not simply due to the combined activity/presence of cohesin and condensin II during mitosis, which also occurs in *Wapl* single mutants. It only arises when both cohesin and condensin were stably associated with chromatin during G2.

We conclude that combining the abnormal activity of condensin II unmasked by deleting *Mcph1* with that of cohesin unmasked by deleting *Wapl* leads to a major transformation of chromosome structure when cells enter G2, that is associated with coiling of the entire axis of the chromosome. Interestingly, this coiling does not have a handedness that persists throughout the chromosome in metaphase. Instead, the chromosome appears divided into segments whose axes are coiled with alternating handedness.

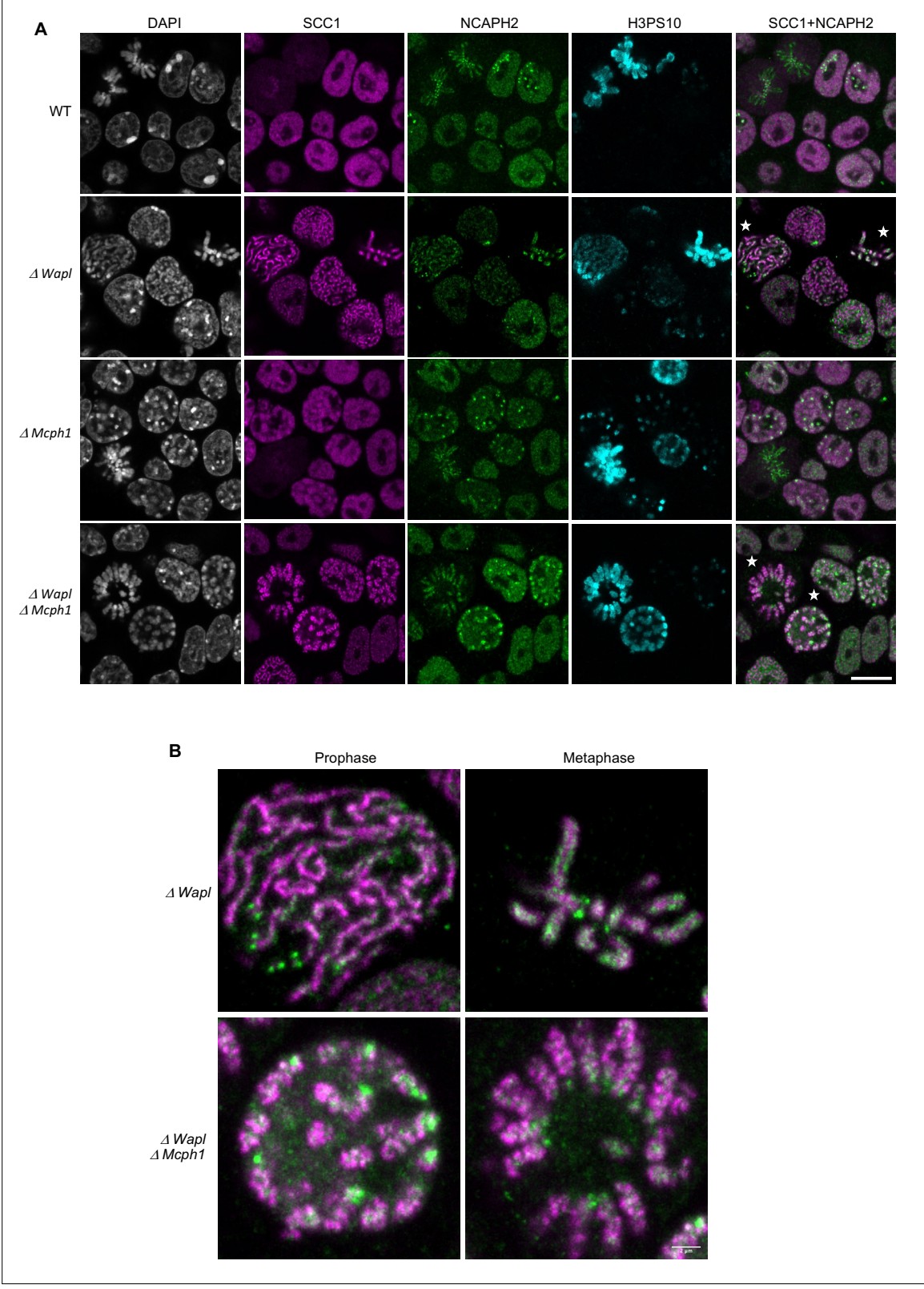

**Figure 10.** *Mcph1 deletion induces the coiling of the vermicelli.* (**A**) Immunofluorescence analysis of the chromatin organisation in the four conditions: wild-type, Δ*Wapl*, Δ*Mcph1*, and Δ*Wapl* Δ*Mcph1*. In order to compare cells that are in G2, prophase or metaphase, Histone H3-serine 10 (cyan) was used as a cell cycle marker. The localisation of SCC1-Halo was analysed using Halo-JFX554, NCAPH2-GFP using nanobodies and DNA was detected using DAPI. (**B**) Magnified view of cells marked with a white star in panel A. Scale bar, 5 µm.

*Figure 10 continued on next page*

*Figure 10 continued*

The online version of this article includes the following figure supplement(s) for figure 10:

**Figure supplement 1.** Western blot analysis of the four conditions analysed in *Figure 10* and *Figure 11*.

**Figure supplement 1—source data 1.** Raw data uncropped blots corresponding to *Figure 10—figure supplement 1*.

**Figure supplement 2.** *Quantification of NCAPH2-GFP and SCC1-Halo on the chromosome axes in ΔWapl cells.*

**Figure supplement 2—source data 1.** Raw data of the fluorescence quantification.

## Discussion

Despite accumulation within interphase nuclei, condensin II associates with chromatin only fleetingly if at all and exerts little or no effect on chromosome topology during this phase of the cell cycle. It normally only associates stably with chromosomal DNA and organises it into a series of loops when cells enter M phase. The restriction of condensin II's activity to M phase was previously attributed to its phosphorylation by CDK1 (*Abe et al., 2011*; *Alderton et al., 2006*; *Gruber et al., 2011*; *Tibelius et al., 2009*). Our finding that in the absence of MCPH1, condensin II is capable of transforming the topology of chromosomal DNA in cells arrested in G2 using a CDK1 inhibitor implies that condensin II is capable of substantial activity in the absence of CDK1-mediated phosphorylation normally associated with M phase. In other words, it is MCPH1 that prevents condensin II's association with chromosomes during G2 not the lack of CDK1 phosphorylation.

Our observation that phosphorylation of a CDK1 consensus sequence abolishes association between a conserved and essential SLiM within MCPH1's central domain and condensin II's NCAPG2 subunit raises the possibility that CDK1 exerts at least part of its effect by preventing MCPH1's association with condensin II. Our conclusion that the SLiM within MCPH1's central domain is essential for its inhibitory activity contradicts the claim that it is not necessary, an inconsistency that we attribute to the fact that previous studies tested the function of over-expressed MCPH1 alleles (*Wood et al., 2008*; *Yamashita et al., 2011*). Accordingly, when a version of *Mcph1* deleted for the SLiM domain was overexpressed in mouse oocyte, it was still inhibiting Condensin II association with chromosomes (data not shown). This result suggests that MCPH1 with a deletion of this short motif can still have an effect on condensin II when expressed at high levels and is consistent with previous studies. However, we have shown that the SLiM is essential for the inhibition of condensin II when the protein is expressed at endogenous levels, demonstrating the importance of performing experiments using endogenous protein levels to prevent the conclusion being biased by the protein expression levels.

Our findings as well as those of others (*Arroyo et al., 2017*; *Neitzel et al., 2002*; *Trimborn et al., 2004*) show that MCPH1 is responsible for inhibiting condensin II during G1 as well as G2 phase. Thus, in the absence of MCPH1, condensin II organises chromosomal DNAs into chromatid-like structures during G1 and G2 but strikingly not during S phase when some other (MCPH1-independent) mechanism prevents it from associating stably with chromatin.

MCPH1's inhibition of condensin II depends on its N-terminal BRCT domain in addition to its central SLiM. Indeed, most of *MCPH1* mutations identified in microcephaly patients affect the BRCT domain, which possibly interacts with some other part of condensin II once MCPH1 has been recruited via its SLiM. However, the function of this domain remains mysterious.

How does MCPH1 inhibit condensin II? An important clue stemmed from the numerous similarities between MCPH1 and WAPL, a protein that facilitates cohesin's release from chromatin (*Yatskevich et al., 2019*). WAPL binds to STAG, the cohesin subunit equivalent to NCAPG2, using a SLiM and its inactivation leads to cohesin's stable association with chromatin (*Li et al., 2020*; *Tedeschi et al., 2013*; *Wutz et al., 2020*). Cohesin release mediated by WAPL involves dissociation of the NTD of cohesin's SCC1 kleisin subunit from the coiled coil that emerges from SMC3's ATPase head, known as its neck (*Beckouët et al., 2016*; *Chan et al., 2012*; *Eichinger et al., 2013*). It is currently thought that kleisin-neck dissociation takes place, albeit rarely, upon engagement of cohesin's SMC1 and SMC3 ATPase heads in the presence of ATP when SMC3 is unacetylated and in the absence of SCC2 (*Beckouët et al., 2016*; *Chan et al., 2013*; *Chan et al., 2012*; *Srinivasan et al., 2019*). Crucially, the fusion of SMC3's C-terminus to SCC1's N-terminus completely blocks WAPL from triggering cohesin's release from chromatin (*Chan et al., 2012*; *Eichinger et al., 2013*). It could do so either by creating a barrier to the passage of DNA through an opened kleisin-neck interface, i.e. by blocking the exit of DNAs

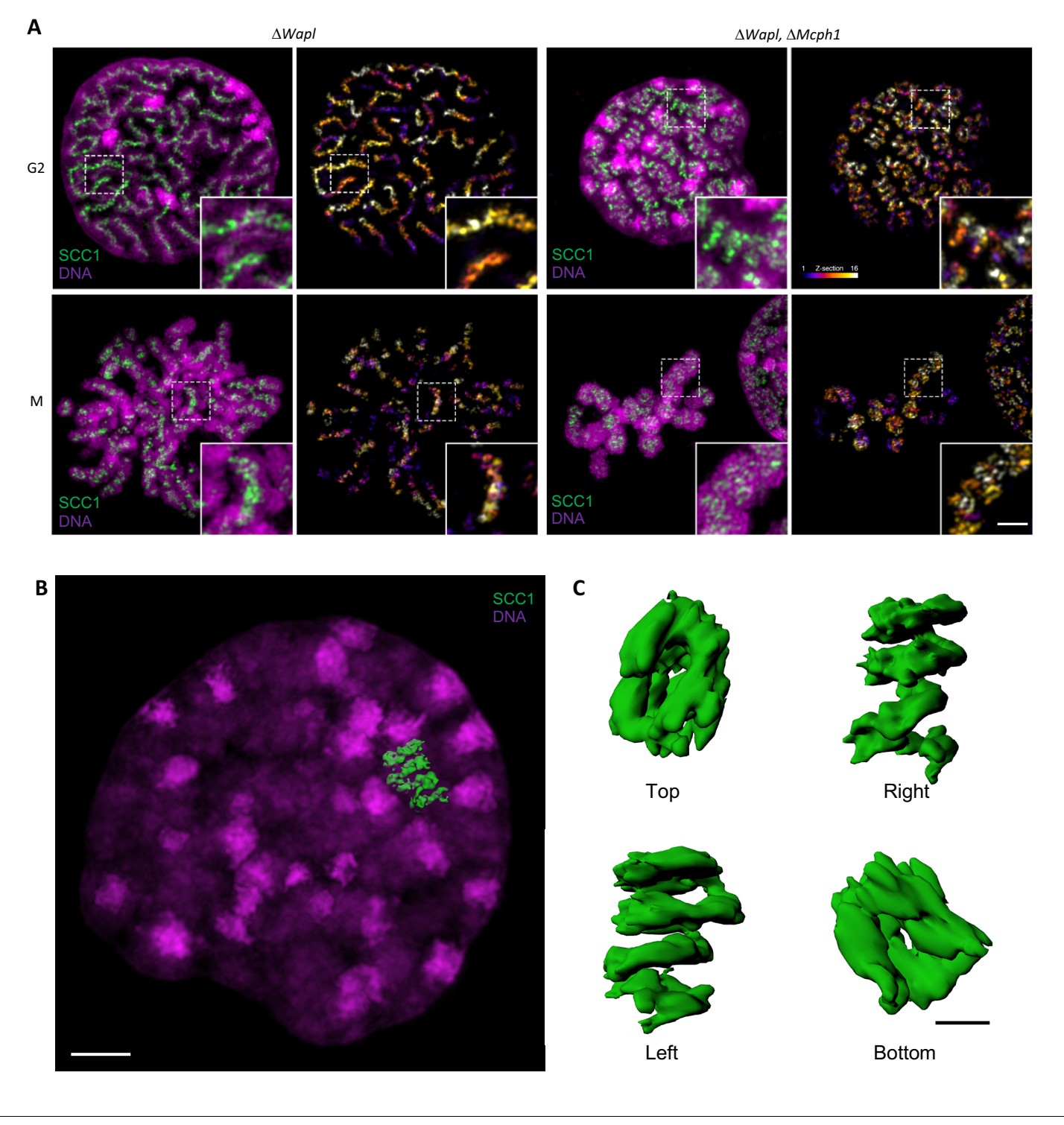

**Figure 11.** *Super-resolution 3D-SIM analysis of G2 and metaphase cells deleted for Wapl or both Mcph1 and Wapl.* (**A**) Cells deleted for Δ*Wapl* or Δ*Wapl*+Δ*Mcph1* in G2 or metaphase (**M**) were analysed by 3D-SIM. Maximum intensity projections of 16 consecutive mid sections covering 2 µm in depth. The left panel shows DNA coloured in magenta and SCC1-Halo in green. The right panel shows the SCC1 signal with z-depth colour-coded. Scale bar, 5 µm (inset, 1 µm) (**B**) Representative SCC1-Halo solenoid structure corresponding to one chromosome from a Δ*Wapl*+Δ*Mcph1* cell in G2 was segmented (green) and overlaid to the DNA (magenta). Scale bar, 2 µm. (**C**) 3D surface rendering of the segmented and isolated solenoid from panel B. View from the top, right, left, and bottom of one segmented solenoid. Scale bar, 1 µm.

The online version of this article includes the following video for figure 11:

*Figure 11 continued on next page*

*Figure 11 continued*

**Figure 11—video 1.** 3D video rendering of the segmented and isolated solenoid from Figure 11 panel B.
https://elifesciences.org/articles/73348/figures#fig11video1

previously entrapped within cohesin rings, or merely by hindering kleisin-neck dissociation, which has some other poorly understood function necessary for release.

Our finding that an analogous fusion, between the C-terminus of SMC2 and the N-terminus of NCAPH2, prevents the release of condensin II from meiosis I chromosomes in oocytes upon over-expression of MCPH1 suggests that MCPH1 prevents condensin II's association with chromatin by a mechanism that is similar to cohesin's release by WAPL. It has been reported that engagement of SMC2 and SMC4 heads triggered by ATP binding induces the release of the NTD of the kleisin from SMC2's neck (*Hassler et al., 2019*), raising the possibility that MCPH1 blocks condensin II's stable association with chromatin by facilitating such a process. Irrespective of the actual mechanics, which remains poorly understood, our observations emphasise that MCPH1 blocks condensin II's association with chromosomes using a mechanism similar to that that employed by WAPL to cause cohesin release.

Despite these striking similarities, the mode of action of these two proteins may differ in an important respect. Though WAPL alters cohesin's residence time on chromatin exclusively by facilitating release, our experiments demonstrated that MCPH1 was unable to release condensin II from chromosomes arrested in meiosis I, suggesting that over-expression of MCPH1 in oocytes prevents de novo association of condensin II but does not remove complexes previously associated with chromosomes (data not shown).

Recent advances have shown that both human condensin and cohesin complexes extrude DNA loops (*Davidson et al., 2019*; *Golfier et al., 2020*; *Kim et al., 2019*; *Kong et al., 2020*), and cryo-electron microscopy structures of yeast condensin, and yeast and human cohesin are starting to provide insight into how these complexes engage with DNA (*Collier et al., 2020*; *Higashi et al., 2020*; *Shi et al., 2020*). In the case of cohesin, it has been suggested that an early step is the clamping of DNA on top of SMC1 and SMC3 ATPase domains that have engaged with each other in the presence of ATP. Clamping in this manner requires cohesin's SCC2 HAWK protein, which is equivalent to condensin II's NCAPD3. SCC2 is necessary to prevent cohesin's release from chromatin and may perform this function by preventing dissociation of its kleisin subunit from SMC3's neck (*Collier et al., 2020*; *Srinivasan et al., 2019*). If NCAPD3 had a similar role, then MCPH1 could conceivably block condensin II's association with chromatin by interfering with NCAPD3's ability to block the release of NCAPH2 from SMC2.

One of the earliest insights into the cell division cycle was that one of the main constituents of the nucleus undergoes a dramatic morphological transformation shortly before division, namely the transformation of an apparently amorphous mass of chromatin into thread-like structures now known as chromosomes, each composed of a pair of chromatids joined together. It is now recognised that this transformation is brought about by condensins I and II which act by extruding DNA loops. It is also recognised that interphase chromosomal DNA is not in fact amorphous but instead characterised by a complex and dynamic network of interactions known as topologically associated domains or TADs, which are created by cohesin that like condensin is a DNA loop extruder, albeit one that is active during interphase and blocked by the site specific DNA-binding protein CTCF (*Li et al., 2020*; *Rao et al., 2017*).

There are two reasons why DNAs are not organised into thread-like chromatids during interphase. By causing cohesin release, WAPL prevents loop extrusion by cohesin going to completion while MCPH1 prevents condensin II associating with DNA stably and thereby extruding loops. Crucially, inactivation of either protein leads to the formation of chromatid-like structures during interphase, albeit by different loop extruders and with different actual morphologies. Interestingly, depletion of both proteins simultaneously leads to a major transformation of chromosome structure as cells enter mitosis, which is associated with coiling of the entire axis of the chromosome. Understanding how and why this comes about through the unregulated activities of cohesin and condensin II may help reveal further insight into how loop extrusion creates chromosomes.

## Materials and methods

### Mouse strain, in vitro culture, and oocytes micro-injection

*Ncaph2*<tm1a(EUCOMM)Wtsi> (MBCH;EPD0070-2-G090) were obtained from the Wellcome Trust Sanger Institute. The corresponding *flox* allele was obtained as previously described (*Houlard et al., 2015*). The fusion was obtained by cloning in frame *Smc2* cDNA, a linker containing three TEV protease cleavage sites (GGGGSGGGGSGGGGTGSENLYFQGPRENLYFQGGGSENLYFQGTRGGGGSGGGGSGGGG), *Ncaph2* cDNA (Origene, MC200537) and the eGFP ORF in the pUC19 vector.

Fully grown prophase-arrested GV oocytes were isolated and injected with mRNAs (5–10 pl) diluted in RNase-free water at the following concentrations: *H2b–mCherry*: 150 ng µl$^{-1}$, *Mad2*: 200 ng µl$^{-1}$, *NCAPH2–Gfp*: 50 ng µl$^{-1}$, *Fusion-EGFP* (50 ng µl$^{-1}$), *Mcph1* (200 ng µl$^{-1}$), *Tev* protease: 250 ng µl$^{-1}$. All experimental procedures were approved by the University of Oxford ethical review committee and licensed by the Home Office under the Animal (Scientific procedures) Act 1986. No statistical method was used to predetermine sample size. The experiments were not randomised and the investigators were not blinded to allocation during experiments and outcome assessment.

### Live cell confocal imaging

Oocyte live cell imaging was done in four-well Labtek chambers (ref. 155383) in 4 µl drops of M16 medium covered with mineral oil (Sigma) in a 5 % $CO_2$ environmental microscope incubator at 37 °C (Pecon). Images were acquired using an LSM-780 confocal microscope (Zeiss) using the ZEN 2011 software. Between 7 and 12 slices (between 1 and 4 µm) were acquired every 5–15 min for each stage position using the autofocus tracking macro developed in J. Ellenberg's laboratory at EMBL. For detection of eGFP and mCherry, 488 nm and 561 nm excitation wavelengths and MBS 488/561 filters were used. Images were further analysed using Volocity software. For high-resolution videos, the lens used was a C-Apochromat ×63/1.20 W Corr UV–VIS–IR. The fluorescence was quantified using Fiji.

### E14 mouse embryonic stem cells culture

ES-E14 mouse embryonic stem cells (RRID:CVCL_C320). These E14 cells grow as colonies in 3D and depends on the presence of Leukemia Inhibiting Factor in the media. They were authenticated by genome sequencing during HiC and ChIP seq experiments. The cells were regularly tested negative for mycoplasma. E14 mouse embryonic stem cells were grown in Dulbecco's modified Eagle's medium (DMEM; Life Technologies) supplemented with 10 % foetal calf serum (Seralab), 2 mM L-glutamine (Life Technologies), 1 x non-essential amino acids (Life Technologies), 50 µM β-mercaptoethanol (Life Technologies), 1 X penicillin-streptomycin solution (Life Technologies) and leukaemia inhibitory factor (LIF) made in-house. All E14 cells were grown in feeder-free conditions on gelatinised plates at 37 °C in a humid atmosphere with 5 % $CO_2$.

For conditional deletion of *Wapl*, Cre recombinase was induced by treating the cells with 800 nM 4-hydroxytamoxifen (OHT) for the indicated time. The degradation of NCAPH2-HALO was triggered by adding HaloTag PROTAC Ligand (Promega) at 1 µM for 16 hours.

The Halo Ligands (Halo-TMR Ligand, Promega, Ref G8251) were added in the culture medium for 20 min at 100 nM. Cells were washed and left in the incubator with fresh medium for an extra 30 min to remove the unbound ligand before being analysed by immunofluorescence or western blot.

### Genomic engineering using CRISPR homology-directed repair

The sgRNAs were designed using the CRISPOR online tool (http://crispor.tefor.net/crispor.py) and cloned in the pSptCas9(BB)–2A-Puro(PX459)-V2.0 vector (Addgene #62988). The sgRNA cloning was done according to the protocol from *Ran et al., 2013*.

To generate the targeting constructs, 1 kb homology arms were amplified by PCR (Q5-NEB) from E14 cells genomic DNA and cloned in pUC19 vector using Gibson Assembly Master Mix kit (New England Biolabs). The targeting construct was designed such that the guide RNA sequence used for the specific targeting was interrupted by the tag or contained silent mutations.

For each targeting, a 6 cm dish of E14 cells 50 % confluent was transfected using 2 µg of pX459-Cas9-sgRNA and 5 µg of targeting construct using Lipofectamine 2000 (ThermoFisher) according to manufacturer's guidelines. The next day, cells were trypsinised and plated at three different densities in 20 cm dishes in medium supplemented with puromycin (1 µg/ml). The selection medium was removed 48 h later and cells were grown for approximately 10 days. Nienty-six Individual clones were

then picked in 96 well plates, grown for 48 hr and split into two 96-well plates. The next day, genomic DNA was prepared using 50 µl of Lysis buffer (10 mM Tris HCl pH8, 1 mM EDTA, 25 mM NaCl, 200 µg/ml Proteinase K), incubated at 65 °C for 1 hr, 95 °C for 10 min to inactivate Proteinase K. The clones were then screened by PCR (Q5-NEB) and amplified to be further analysed by western blot. The deletion of MCPH1 central domain was confirmed by sequencing.

## Conditional *Wapl* deletion

To generate the TEV protease conditional cells, a series of four tandem STOP cassette (Addgene, pBS.DAT-LoxStop, Jacks Lab) flanked by two LoxP sites was cloned between the pCAG promoter and the PK tagged Tev protease cDNA. The CRE-ERT2 cDNA was cloned upstream under the transcriptional control of the Rosa26 Splice acceptor. This construct was then flanked by 1 kb homology arms to target the construct at the Rosa26 locus using CRISPR-HDR. After selecting and amplifying of the targeted E14 clones, the TEV protease could be detected by western blot 8 hr post hydroxytamoxifen induction. Immunofluorescence analysis revealed a homogeneous expression of TEV protease in all the cells.

In the selected clones, three TEV protease cleavage sites were targeted in Wapl coding sequence after Proline 499 using CRISPR-HDR. After selecting of the targeted clones, it appeared that TEV cleavage of the Wapl protein led to a new steady-state in the cells in which a small amount of full-length protein was present preventing the formation of vermicelli. To avoid this compensation effect, we targeted loxP sites in the *Wapl* gene on both sides of exon four to induce the deletion of the gene simultaneously as the TEV cleavage of the protein already present in the cell. This combined strategy induced a complete loss of function of *Wapl* within 8 hr and the formation of vermicelli.

## sgRNA used for CRISPR-HDR

| Targeted Gene | sgRNA |
|---|---|
| *Ncaph2-Gfp* | GGTGGAAAGTAGTATATACC |
| *Ncaph2-Halo* | GGTGGAAAGTAGTATATACC |
| *Mcph1 deletion Sg5'* | GGTGTGCAATTCCTAGTGTG |
| *Mcph1 deletion Sg3'* | AGCTGTTCCTTAGAACACGA |
| *Mcph1-Gfp* | ACAGTGAGACATCTACAATG |
| *Stop-Tev* | CATGGATTTCTCCGGTGAAT |
| *Wapl-Lox-Tev Sg5'* | AATGGGTGCTTATAATTAGC |
| *Wapl-Lox-Tev Sg3'* | ACAATGTCACAATGGCTCAT |
| *Scc1-Halo* | ATAATATGGAACCGTGGTCC |
| *DelCen-Mcph1* | cgttgaggcttcttcctatg |
| TEV sites in Wapl Two Guide RNA used in combination | AATGGGTGCTTATAATTAGC<br>ACAATGTCACAATGGCTCAT |

## Immunofluorescence detection

E14 cells were plated on glass coverslips (Marienfeld, High precision, 22 × 22, N°1.5H, Ref 0107052) in six-well plates. Forty-eight hr later, cells were labelled with the Halo Ligand (Halo-JFX554, Janelia 100 nM 30 min) (*Grimm et al., 2021*) or EdU (5 min at 10 µM), washed PBS and then fixed in 3.8 % Formaldehyde (Sigma-F8775) in PBS for 15 min. After three PBS washes, cells were permeabilised in 0.5 % Triton X-100 in PBS for 10 min and then washed three times in PBS. After 20 min in blocking buffer: PBS 3 % BSA (Sigma, A4503). The coverslips were transferred in a wet chamber and covered with 100 µl of antibody solution in blocking buffer. After 1 hr at room temperature, the coverslips were washed three times in blocking solution and incubated with the secondary antibody for 1 hr

(AlexaFluor 594 or 488, 1/500, Life Technology). After three washes in PBS, DNA was labelled using Hoechst (Sigma, 33342, 5 µg/ml) for 15 min. The coverslips were then washed in PBS, mounted in Vectashield (H1000) and sealed using nail varnish. Imaging was done using an LSM 780 confocal microscope (Zeiss) using the ZEN 2011 software. EdU was detected according to the manufacturer's instruction (Click-iT EdU Imaging kit, Invitrogen). CDK1 inhibition was performed by incubating the cells in E14 medium supplemented with 9 µM RO-3306 (SIGMA SML0569) for 7 hr before fixation. All experiments based on immunofluorescence where repeated at least three times.

## Immunoprecipitation and western blot

The cells were collected using Trypsin and washed in PBS twice. The cell pellet was resuspended in 10 vol. of buffer A (10 mM HEPES pH7.9, 1.5 mM MgCl2, 10 mM KCl, 1 mM DTT, 1 mM PMSF, 1 x complete protease inhibitor (Roche, 04693132001)) and incubated 15 min on ice. After centrifugation at 4°C at 2000 rpm for 5 min to remove the supernatant, the pellet was resuspended in 3 vol. of buffer A + NP40 (0.1%) and incubated for 15 min on ice followed by another centrifugation at 4 °C 4000 rpm for 5 min to remove the supernatant and resuspended in lysis buffer: (20 mM Tris pH7.5, 200 mM NaCl, 1 % Triton X-100, 0.1 % sodium deoxycholate, 1 mM DTT, 1 mM PMSF, 1 x complete protease inhibitor (Roche, 04693132001), 1 x super nuclease). The suspension was passed through a 26GA needle 10 times, left on ice for 1 hr and centrifuged at 20,000 g for 30 min. Proteins were quantified by Bradford (KitBiorad 5000006, spectrophotometer BIOCHROM). Each IP was performed using 1 mg of protein in 1 ml in lysis buffer. GFP-Trap agarose beads (GTA-10, Chromotek) were washed in lysis buffer and incubated according to the supplier's instructions with the protein extract overnight at 4 °C. Washed five times in Lysis buffer and separated by SDS-PAGE, imaged using Fujifilm FLA-7000 imager if Halo ligand was used and then transferred overnight on Nitrocellulose membrane. After Ponceau red evaluation of the transfer quality, the membrane was blocked in PBS-0,1%Tween 20 + 5 % non-fatty milk for an hour and incubated with the antibody overnight in PBS-0,1%Tween 20 + 5 % non-fatty milk. Western blots were analysed using LI-COR Odyssey Fc imager.

Immunoprecipitations experiments where repeated three times or more in independent experiments.

## Antibodies list

NCAPH2: Rabbit polyclonal produced on demand by Eurogentec (WB:1/1000).
SCC1: Millipore, 53A303, mouse monoclonal antibody (WB:1/1000).
SMC2: Cell Signaling Technology, D23C5, rabbit monoclonal Antibody (WB:1/500).
Lamin B1: Abcam Ab133741, rabbit monoclonal (WB:1/1000).
MCPH1: Cell Signaling Technology, D38G5, rabbit monoclonal Antibody (WB:1/1000).
CREST: Immunovision HCT0-100, human autoantibody (IF: 1/500)
H3PS10: Millipore, clone 3H10, mouse monoclonal (WB:,1/1000 IF:1/2000).
γH2aX: Millipore, clone JBW301, mouse monoclonal (WB:,1/1000 IF:1/500).
CDK1: Cell Signaling Technology, cdc2, 77055, rabbit polyclonal (WB:1/1000).
Phospho-CDK1: Cell Signaling Technology, phospho-cdc2 (Tyr15) antibody, 9111, rabbit polyclonal (WB:1/1000).
WAPL: provided by J.M. Peters's Lab (WB:1/1000).
GFP: Abcam, ab290, rabbit polyclonal (WB: 1/1000, IF:1/500).
PK-Tag: Biorad, MCA 1360 G, mouse monoclonal (WB:1/1000).
Cyclin B1: Cell Signaling Technology, 4138T, rabbit monoclonal (IF:1/200).
NCAPD3: Bethyl Laboratory, A300-604A-M, rabbit polyclonal (WB:1/1000).

## Halo in gel imaging

Cells were incubated with Halo-TMR ligand (100 nM) for 30 min then washed in ligand-free medium and analysed by immunofluorescence or for protein purification. After SDS-PAGE, the fluorescence was measured in the gel using an Imager Fujifilm FLA-7000.

## FRAP

Live-cell imaging was performed in a spinning disk confocal system (PerkinElmer UltraVIEW) with an EMCCD (Hamamatsu) mounted on an Olympus IX8 microscope with Olympus 60 × 1.4 N.A. and 100

× 1.35 N.A. objectives. Image acquisition and quantitation were performed using Volocity software. During imaging, cells were maintained at 37 °C and 5 % CO2 in a humidified chamber. FRAP was carried out with a 488 nm laser beam, 100 % power, 15–30 ms. The fluorescence intensity measurement was performed by using ImageJ. All signals were subjected to background correction. The fluorescence intensity of unbleached and bleached areas was normalised to that of initial pre-bleaching images using the EasyFRAP website.

## In-situ Hi-C

Hi-C libraries were generated as described in *Rao et al., 2014*, analysed using the Juicer pipeline (*Durand et al., 2016b*) and visualised with Juicebox (*Durand et al., 2016a*). We sequenced 785,454,085 Hi-C read pairs in wild-type mouse ES cells, yielding 509,278,039 Hi-C contacts; we also sequenced 1,814,815,287 Hi-C read pairs in *Mcph1*-deleted cells, yielding 1,284,169,272 Hi-C contacts. Loci were assigned to A and B compartments at 100 kB resolution. Loops were called with HiCCUPS at 25 kB, 10 kB, and 5 kB resolution. Contact domains were called at 10 kB resolution. Contact frequency analysis was performed as described in *Sanborn et al., 2015*. All code used for these analyses is publicly available at https://github.com/aidenlab/juicer (analysis software, building hic file)(*Aiden Lab, 2020a*), https://github.com/aidenlab/juicebox (visualization software)(*Aiden Lab, 2021a*), https://github.com/aidenlab/straw (stream hic file data into python for custom analysis) (*Aiden Lab, 2021c*), https://github.com/aidenlab/contact-probability (contact probability analysis) copy archived at swh:1:rev:d5485dbdf555df3fb540bb907e401b8f26252f2c (*Aiden Lab, 2020b*) and https://github.com/aidenlab/mcph1_notebook (AB enrichment analysis/plotting) copy archieved at swh:1:rev:1e46e50a2c9e1e1d141311a27c190ea27bd4ef9a (*Aiden Lab, 2021b*). With a help forum for questions at (aidenlab.org/forum.html).

## Structured illumination microscopy

Super-resolution 3D-SIM was performed on a DeltaVision OMX SR system (GE Healthcare) equipped with sCMOS cameras (PCO) and 405, 488 and 568 nm lasers, using a 60 x NA 1.42 PlanApo oil immersion objective (Olympus). To minimise artefacts due to spherical aberration.

Raw data sets were acquired with a z-distance of 125 nm and 15 raw images per plane (five phases, 3 angles). Reconstructions were performed with SoftWoRx 6.2 (GE Healthcare) using channel-specifically measured optical transfer functions (OTFs) generated from 100 nm diameter green and red FluoSphere beads (ThermoFisher), respectively, and Wiener filter set to 0.0030. For DAPI acquisitions, the sample was excited with the 405 nm laser and the emission detected in the green channel and reconstructed with a green OTF. This is enabled by the broad emission spectrum of DAPI and empirically resulted in better reconstructions than reconstructing blue emission with a 'blue' OTF obtained typically less intense blue FluoSphere beads.

All data underwent quality assessment via SIMcheck (*Ball et al., 2015*) to determine image quality via analysis of modulation contrast to noise ratio (MCNR), spherical aberration mismatch, reconstructed Fourier plot and reconstructed intensity histogram values. Reconstructed 32-bit 3D-SIM datasets were thresholded to the stack modal intensity value and converted to 16-bit composite z-stacks to discard negative intensity values using SIMcheck's 'threshold and 16-bit conversion' utility and MCNR maps were generated using the 'raw data modulation contrast' tool of SIMcheck. To eliminate false positive signals from reconstructed noise, we applied SIMcheck's 'modulation contrast filter' utility. Briefly, this filter sets masks out all pixels, where the underlying MCNR value in the raw data fall below an empirically chosen threshold MCNR value of 6.0, followed by a Gaussian filter with 0.8 pixel radius (xy) to smoothen hard edges (*Rodermund et al., 2021*).

Colour channels were registered in 3D with the open-source software Chromagnon 0.85 (*Matsuda et al., 2018*) determining alignment parameter (x,y,z-translation, x,y,z-magnification, and z-rotation) from a 3D-SIM dataset acquired on the date of image acquisition of multicolour-detected 5-ethynyl-2'-deoxyuridine (EdU) pulse replication labelled C127 mouse cells serving as biological 3D alignment calibration sample (*Kraus et al., 2017*).

## FACS

Cells were incubated with EdU (10 µM) for 5 min, collected using trypsin, washed with PBS and fixed in 3.8 % Formaldehyde (Sigma-F8775) in PBS for 15 min. After three PBS washes, cells were

permeabilised in 0.5 % Triton X-100 in PBS for 10 min and then washed three times in PBS. Cells were then incubated for 20 min in blocking buffer: PBS 3 % BSA (Sigma, A4503), incubated for 1 hr with H3PS10 antibody (1/1000), washed three times in PBS and then incubated with fluorescent secondary antibody (Alexa Fluor 488, 1/500, Life Technology) incubated with propidium iodide (Sigma, 30 µg/ml) and then analysed by FACS. 15,000 cells were analysed for each data point. EdU detection was done following the manufacturer's instruction (Click-iT EdU Imaging kit, Invitrogen).

## Protein purification

Human condensin II pentameric and tetrameric complexes were purified as previously described (*Kong et al., 2020*). Full-length *MCPH1* was cloned into the pLIB vector and viral bacmids were generated using Tn7 transposition in DH10EMBacY cells (Geneva biotech), transfected into Sf9 cells using Cellfectin II (GIBCO) and resultant virus harvested after 3 days. Virus was further amplified in Sf9 cells before being used to infect High Five cells for protein expression. High Five cells were harvested by centrifugation 3 days after infection. Cell pellets were resuspended in purification buffer (20 mM HEPES pH 8, 300 mM KCl, 5 mM MgCl$_2$, 1 mM DTT, 10 % glycerol) supplemented with 1 Pierce protease inhibitor EDTA-free tablet (Thermo Scientific) per 50 mL and 25 U/ml of Benzonase (Sigma) and lysed with a Dounce homogenizer followed by brief sonication. The lysate was cleared with centrifugation, loaded onto a StrepTrap HP (GE), washed with purification buffer and eluted with purification buffer supplemented with 5 mM desthiobiotin (Sigma). Size exclusion chromatography was performed using purification buffer on a Superdex 200 16/60 column (GE), and protein containing fractions separated from the void volume were pooled and concentrated.

MCPH1 residue 1–435, 1–195 and 196–435 were cloned into a pET vector with an N-terminal 6xHis-MBP fusion tag or with an N-terminal 6xHis and C-terminal MBP tag, and expressed in *E. coli* BL21 (DE3) pLysS cells (Novagen). Cells were grown at 37 °C, induced with 1 mM IPTG for 4 hr, before being harvested by centrifugation and flash frozen. The cell pellet was resuspended in MCPH1 purification buffer (20 mM Tris pH 7.5, 150 mM NaCl, 10 % glycerol, 1 mM DTT, Pierce protease inhibitor EDTA-free tablet), lysed with sonication on ice, treated with Benzonase (Sigma-Aldrich) (10 µL per 100 mL, with 1 mM MgCl$_2$) and cleared via centrifugation. Cleared cell lysate was filtered with a 5 µm filter, imidazole was added to a final concentration of 10 mM and incubated with pre-equilibrated His-Pure NTA resin. The resin was washed with MCPH1 purification buffer, then washed with wash buffer 20 mM Tris pH 7.5, 500 mM NaCl, 20 mM imidazole, before elution with 20 mM Tris pH 8, 300 mM NaCl, 500 mM Immidazole, 10 % glycerol. Protein was diluted 2-fold with buffer TA (20 mM Tris pH 8, 5 % glycerol, 1 mM DTT), and loaded onto a HiTrap Q Fast Flow column or loaded on to HiTrap Heparin HP column (GE) and eluted with a gradient of buffer TB (20 mM Tris pH 8, 2 M NaCl, 5 % glycerol, 1 mM DTT). For protein used in ATPase assays, final size exclusion chromatography was performed using purification buffer and a Superdex 200 10/300 or 16/60 column (GE).

## Pull-downs

Condensin complexes (0.1 µM) were mixed with at least a 10-fold molecular excess of MBP MCPH1 constructs or MBP protein in 200 µL and incubated with 40 µL of Strep-tactin sepharose resin (IBA) in MCPH1 purification buffer. The resin was washed five times, before being eluted by boiling in 1 x NuPAGE LDS sample buffer with 50 mM DTT. Samples of 10 % input and resin elution were run on 4–12% NuPAGE Bis-Tris gels against Color Prestained Protein Standard, Broad Range (NEB) and stained with Instant Blue (Expedeon) or silver stain (Life Technology).

The absence of the CAP-D3 subunit was confirmed by western blot analysis of the output protein samples. For this, the tetrameric condensin II pull-down was run on an SDS page gel and transferred to a nitrocellulose membrane (Amersham). The membrane was then blocked with 5 % milk-powder

**Table 1.** Peptides used in FP experiments.

| Name | Sequence |
|---|---|
| 5FAM-MCPH1 | 5FAM- CGESSYDDYFSPDNLKER |
| MCPH1 | CGESSYDDYFSPDNLKER |
| MCPH1pS417 | CGESSYDDYF{pSER}PDNLKER |

in TBS-T, before being probed with a mouse anti-CAP-D3 antibody (Santa Cruz, Sc-81597, used at 1/1000), then a goat DyLight 800 florescent anti-mouse secondary antibodies (Cell Signaling Technology, 5257, used at 1/5000). The membrane was imaged using a LI-COR imager.

## Fluorescence polarisation assays

Peptides used in fluorescence polarisation assays were synthesised by Genscript and are shown in *Table 1*. The concentration of 5FAM wild-type MCPH1$_{407-422}$ was determined using the 5-FAM extinction coefficient of 83,000 $(cmM)^{-1}$ at 493 nm. Non-labelled peptides had TFA removed to less than 1 % and were accurately quantified using Genscript's amino acid analysis service. All peptides were solubilised in DMSO and diluted to a working concentration in FP assay buffer (20 mM Tris pH 7.5, 200 mM NaCl, 1 mM DTT). Peptides in competition experiments were diluted in a two-fold series with FP assay buffer supplemented with DMSO, such that DMSO concentration was constant for all peptide concentrations.

FP binding assays were performed with 0.3 µM of 5-FAM-labelled wild-type MCPH1$_{407-422}$ and 0.625, 1.25, 1.75, 2.5, 3.5 and 5 µM of pentameric condensin II or tetrameric condensin II lacking the CAPG2 subunit, in a total volume of 40 µl in half-area black plates (Constar). The plate was incubated at room temperature for 20 min, before being read with an Omega plate reader (BMG Labtech) at 5 min intervals and monitored to ensure binding had reached equilibrium. Each plate was read three times, and three replicates were performed at each protein concentration.

FP competition titrations were performed as above, but with 0.63 µM of condensin II, 0.3 µM of 5FAM-MCPH1$_{407-422}$ and indicated amount of competing peptide. Data was normalised by subtracting the FP signal for 5FAM-MCPH1$_{407-422}$ in the absence of condensin II at each peptide concentration and divided by the background-subtracted signal of sample with 0.63 µM of condensin II and 0.3 µM of 5FAM-MCPH1$_{407-422}$ without any competing peptide. Each plate was read four times and each experiment was performed a total of three times.

Fluorescence polarisation data was fit using equations for direct binding and directly competitive binding, as presented by *Roehrl et al., 2004*.

## ATPase assays

Complexes of wild type or ATPase hydrolysis deficient Q-loop mutants condensin II with MCPH1$_{1-435}$MBP were purified with gel filtration on a Superose 6 10/300 column, along with wild-type only control in ATPase assay buffer (20 mM Tris pH 7.5, 150 mM NaCl, 1 mM DTT). Prior to gel-filtration all samples were treated with TEV protease to remove N-terminal 6 x His-tag on MCPH1$_{1-435}$MBP. ATPase assays were performed using the EnzChek Phosphate Assay Kit (Invitrogen) modified for a 96 well plate format (*Voulgaris and Gligoris, 2019*). 50 bp double-stranded DNA sequence with the same as that used in the EMSA (without a fluorescent label). Reactions contained 50 nM protein with or without 800 nM DNA. Final conditions included 1 mM ATP and a total salt concentration of 50 mM. Protein/DNA was preincubated in reaction mix without ATP for 15 min at room temperature before the reaction was started by addition of ATP immediately before putting it in the plate reader to track phosphate release. A standard curve ATPase rate was determined from a linear fit of data from the first 60 min.

## Electromobility shift assays

EMSAs were performed using 50 nM of 50 bp double-stranded DNA labelled with Cy5 at the 5' with the sequence:

CTGTCACACCCTGTCACACCCTGTCACACCCTGTCACACCCTGTCACACC

For MCPH1 EMSAs, MCPH1-MBP constructs were diluted in 20 mM Tris pH 7.5, 150 mM NaCl, 10 % glycerol, 1 mM DTT and mixed 1 in two with DNA in 20 mM HEPES pH 8, 300 mM KCl, 5 mM MgCl$_2$, 10 % glycerol and 1 mM DTT.

For condensin II/MCPH1 EMSAs, condensin II diluted in 20 mM HEPES pH 8, 300 mM KCl, 5 mM MgCl$_2$, 10 % glycerol and 1 mM DTT and mixed with MCPH1$_{1-435}$-MBP, MBP or 5-FAM-MCPH1$_{407-424}$ diluted in in 20 mM Tris pH 7.5, 150 mM NaCl, 10 % glycerol, 1 mM DTT. DNA was then added to a final concentration of 50 nM. Protein and DNA were incubated on ice for 15 min, before being loaded in a 2 % agarose gel and run in 0.5 x TBE buffer for 30 min. Gel was imaged using a Typhoon.

## Acknowledgements

We thank N Halidi and C Monico for technical assistance, M Inês Baptista for her advice on cloning the STOP cassette, A Szczurek for his help in fluorescence quantification, S Mahara for FACS analysis, M Ranes and S Guettler for the MBP vector and N Davey for SLiM conversations. This work was supported by the Wellcome Trust (Grant Ref 107935/Z/15/Z), ERC grant (Proposal No 294401) and Cancer Research UK (26747) to KN. This work was also funded by the Cancer Research UK Programme Foundation (CR-UK C47547/A21536) and a Wellcome Trust Investigator Award (200818/Z/16/Z) to A.V. Imaging was performed at the Micron Oxford Advanced Bioimaging Unit funded by a Wellcome Trust Strategic Award (091911 and 107457/Z/15/Z). Funding for MSS was provided by the Paul and Daisy Soros Foundation. E.L.A. was supported by the Welch Foundation (Q-1866), a McNair Medical Institute Scholar Award, an NIH Encyclopedia of DNA Elements Mapping Center Award (UM1HG009375), a US-Israel Binational Science Foundation Award (2019276), the Behavioral Plasticity Research Institute (NSF DBI-2021795), NSF Physics Frontiers Center Award (NSF PHY-2019745), and an NIH CEGS Award (RM1HG011016-01A1).

## Additional information

### Funding

| Funder | Grant reference number | Author |
| --- | --- | --- |
| Wellcome Trust | 107935/Z/15/Z | Martin Houlard<br>Jonathan Godwin<br>Kim Nasmyth |
| European Research Council | 294401 | Martin Houlard<br>Jonathan Godwin<br>Kim Nasmyth |
| Cancer Research UK | 26747 | Martin Houlard<br>Jonathan Godwin<br>Kim Nasmyth |
| Wellcome Trust | 107457/Z/15/Z | Lothar Schermelleh |
| Wellcome Trust | 091911 | Lothar Schermelleh |
| Paul and Daisy Soros Foundation | | Muhammad S Shamim |
| Cancer Research UK | CR-UK C47547/A21536 | Erin E Cutts<br>Alessandro Vannini |
| Wellcome Trust | 200818/Z/16/Z | Erin E Cutts<br>Alessandro Vannini |
| Welch Foundation | Q-1866 | David Weisz<br>Aviva Presser Aiden<br>Erez Lieberman-Aiden |
| McNair Medical Institute Scholar Award | | David Weisz<br>Aviva Presser Aiden<br>Erez Lieberman-Aiden |
| NIH Encyclopedia of DNA Elements Mapping Center Award | UM1HG009375 | Muhammad S Shamim<br>David Weisz<br>Aviva Presser Aiden<br>Erez Lieberman-Aiden |
| US-Israel Binational Science Foundation Award | 2019276 | David Weisz<br>Aviva Presser Aiden<br>Erez Lieberman-Aiden |
| Behavioral Plasticity Research Institute | NSF DBI-2021795 | David Weisz<br>Aviva Presser Aiden<br>Erez Lieberman-Aiden |

| Funder | Grant reference number | Author |
| --- | --- | --- |
| NSF Physics Frontiers Center Award | NSF PHY-2019745 | Muhammad S Shamim<br>Aviva Presser Aiden<br>Erez Lieberman-Aiden<br>Lothar Schermelleh |
| NIH CEGS Award | RM1HG011016-01A1 | David Weisz<br>Aviva Presser Aiden<br>Erez Lieberman-Aiden |

The funders had no role in study design, data collection and interpretation, or the decision to submit the work for publication.

## Author contributions

Martin Houlard, Conceptualization, Data curation, Formal analysis, Investigation, Methodology, Validation, Writing – original draft; Erin E Cutts, Conceptualization, Data curation, Formal analysis, Investigation, Methodology, Software, Validation, Writing – original draft; Muhammad S Shamim, Data curation, Formal analysis, Validation, Visualization; Jonathan Godwin, Methodology, Resources; David Weisz, Formal analysis, Software; Aviva Presser Aiden, Formal analysis, Resources, Validation; Erez Lieberman Aiden, Data curation, Formal analysis, Funding acquisition, Methodology, Resources, Software, Supervision, Validation; Lothar Schermelleh, Data curation, Formal analysis, Methodology, Software, Visualization, Writing – review and editing; Alessandro Vannini, Data curation, Funding acquisition, Methodology, Supervision, Validation, Writing – original draft, Writing – review and editing; Kim Nasmyth, Conceptualization, Funding acquisition, Project administration, Resources, Supervision, Writing – original draft, Writing – review and editing

## Author ORCIDs

Martin Houlard ![ORCID] http://orcid.org/0000-0001-6780-6939
Erin E Cutts ![ORCID] http://orcid.org/0000-0003-3290-4293
Lothar Schermelleh ![ORCID] http://orcid.org/0000-0002-1612-9699
Alessandro Vannini ![ORCID] http://orcid.org/0000-0001-7212-5425

## Ethics

This study was performed in strict accordance with the recommendations in the Guide for the Care and Use of Laboratory Animals of the National Institutes of Health. All of the animals were handled according to approved institutional animal care and use committee (IACUC) protocols (#08-133) of the University of Arizona. The protocol was approved by the Committee on the Ethics of Animal Experiments of the University of Minnesota (Permit Number: 27-2956). All surgery was performed under sodium pentobarbital anesthesia, and every effort was made to minimize suffering.

## Decision letter and Author response

Decision letter https://doi.org/10.7554/eLife.73348.sa1
Author response https://doi.org/10.7554/eLife.73348.sa2

# Additional files

## Supplementary files

• Transparent reporting form

## Data availability

HiC sequencing data has been deposited in GEO. (accession number: GSE188988).

The following dataset was generated:

| Author(s) | Year | Dataset title | Dataset URL | Database and Identifier |
|---|---|---|---|---|
| Houlard M, Cutts EE, Shamim MS, Godwin J, Weisz D, Schermelleh L, Aiden AP, Aiden EL, Vannini A, Nasmyth K | 2021 | MCPH1 inhibits condensin II during interphase by regulating its SMC2-kleisin interface | https://www.ncbi.nlm.nih.gov/geo/query/acc.cgi?acc=GSE188988 | NCBI Gene Expression Omnibus, GSE188988 |

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

# Appendix 1

## Appendix 1—key resources table

| Reagent type (species) or resource | Designation | Source or reference | Identifiers | Additional information |
|---|---|---|---|---|
| strain, strain background (*Escherichia coli*) | XL1-Blue Competent Cells | Thermo-competent | Prepared in the lab | Used to prepare plasmid DNA |
| Strain, strain background (*S. frugiperda*) | Sf9 insect cells | Thermo Fisher | Cat# 11496015 | |
| Strain, strain background (*Trichoplusia ni*) | High-Five insect cells | Thermo Fisher | Cat# B85502 | |
| strain, strain background (*Escherichia coli*) | DH10EMBacY | Geneva biotech | | |
| genetic reagent (Mouse) | *Ncaph2*<sup>tm1a(EUCOMM)Wtsi</sup>, ZP3-Cre | *Houlard et al., 2015* DOI:10.1038/ncb3167 | | Isolation of Mouse oocytes deleted for Ncaph2 |
| cell line (Mouse) | ES-E14 mouse embryonic stem cells | https://web.expasy.org/ cellosaurus/CVCL_C320 | RRID:CVCL_C320 | |
| cell line (Mouse) | E14-*Ncaph2*<sup>Halo/Halo</sup> | This paper | | E14 cells homozygous Halo tag Cterminal NCAPH2 |
| cell line (Mouse) | E14-*Ncaph2*<sup>Halo/Halo,</sup> *Mcph1*<sup>Δ/Δ</sup> | This paper | | E14 cells homozygous Halo tag C-terminal NCAPH2, deleted for *Mcph1* |
| cell line (Mouse) | E14-*Ncaph2*<sup>GFP/GFP</sup> | This paper | | E14 cells homozygous GFP tag C-terminal NCAPH2 |
| cell line (Mouse) | E14-*Ncaph2*<sup>GFP/GFP</sup> *Mcph1*<sup>Δ</sup> | This paper | | E14 cells homozygous GFP tag C-terminal NCAPH2, deleted for *Mcph1* |
| cell line (Mouse) | E14- *Ncaph2*<sup>Halo/Halo</sup> *Mcph1*<sup>GFP/GFP</sup> | This paper | | E14 cells homozygous GFP tag C-terminal MCPH1 and Halo tag NCAPH2, |
| cell line (Mouse) | E14- *Ncaph2*<sup>Halo/Halo</sup> *Mcph1*<sup>ΔCENGFP/ΔCENGFP</sup> | This paper | | E14 cells homozygous GFP tag C-terminal MCPH1 deleted of central domain,and Halo tag NCAPH2, |
| cell line (Mouse) | E14-SCC1<sup>Halo/Halo</sup>, *Ncaph2*<sup>GFP/GFP</sup>*Wapl*<sup>Lox/Lox,</sup> *Rosa26*<sup>CreERT2-LoxSTOPTEV/CreERT2-LoxSTOPTEV</sup> | This paper | | See *Figure 10* |
| cell line (Mouse) | E14-SCC1<sup>Halo/Halo</sup>, *Ncaph2*<sup>GFP/GFP</sup>*Wapl*<sup>Lox/Lox,</sup> *Rosa26*<sup>CreERT2-LoxSTOPTEV/CreERT2-LoxSTOPTEV</sup>*Mcph1*<sup>Δ/Δ</sup> | This paper | | See *Figure 10* |
| transfected construct (mouse) | pUC19-NCAPH2 Halo TV | This paper | | Targeting Halo tag C-terminal Ncaph2 |
| transfected construct (mouse) | pUC19-NCAPH2 GFP TV | This paper | | Targeting GFP tag C-terminal Ncaph2 |
| transfected construct (mouse) | pUC19-MCPH1 GFP TV | This paper | | Targeting GFP tag C-terminal MCPH1 |
| transfected construct (mouse) | pUC19-MCPH1 ΔCEN TV | This paper | | Targeting deletion of the central domain of MCPH1 |
| transfected construct (mouse) | pUC19-SCC1 Halo TV | This paper | | Targeting Halo tag C-terminal SCC1 |

*Appendix 1 Continued on next page*

| Reagent type (species) or resource | Designation | Source or reference | Identifiers | Additional information |
|---|---|---|---|---|
| transfected construct (mouse) | pUC19-Rosa26 STOPLoxTEV TV | This paper | | Targeting the insertion of the STOP-Lox-TEV cassette at Rosa 26 |
| transfected construct (mouse) | pUC19-WAPL TEV LOX TV | This paper | | Targeting the TEV sites and the LoxP sites in *Wapl* |
| antibody | NCAPH2 (rabbit polyclonal) | Produced on demand by Eurogentec | | WB: (1/1000) |
| antibody | SCC1 (mouse monoclonal) | Millipore | 53 A303 | WB: (1/1000) |
| antibody | SMC2 (rabbit monoclonal) | Cell Signaling Technology | D23C5 | WB: (1/500) |
| antibody | Lamin B1 (rabbit monoclonal) | Abcam | Ab133741 | WB: (1/1000) |
| antibody | MCPH1 (rabbit monoclonal) | Cell Signaling Technology | D38G5 | WB: (1/1000) |
| antibody | CREST (human autoantibody) | Immunovision | HCTO-100 | IF: (1/500) |
| antibody | H3PS10 (mouse monoclonal) | Millipore | 3 H10 | WB: (1/1000) IF: (1/2000) |
| antibody | ΔH2AX (mouse monoclonal) | Millipore | JBW301 | WB: (1/1000) IF: (1/500) |
| antibody | CDK1 (rabbit polyclonal) | Cell Signaling Technology | 77,055 | WB: (1/1000) |
| antibody | Phospho-CDK1 (rabbit polyclonal) | Cell Signaling Technology | 9,111 | WB: (1/1000) |
| antibody | WAPL (rabbit polyclonal) | J.M. Peters Lab | | WB: (1/1000) |
| antibody | GFP (rabbit polyclonal) | Abcam | Ab290 | WB: (1/1000) IF: (1/500) |
| antibody | PK-Tag (mouse monoclonal) | Biorad | MCA 1360 G | WB: (1/1000) |
| antibody | Cyclin B1 (rabbit monoclonal) | Cell Signaling Technology | 4138T | IF: (1/200) |
| antibody | NCAPD3 (rabbit polyclonal) | Bethyl Laboratories | A300-604A-M | WB: (1/1000) |
| antibody | AlexaFluor-488 secondary antibody anti mouse | Life Technology | A-11001 | Secondary antibody for IF |
| antibody | AlexaFluor-488 secondary antibody anti rabbit | Life Technology | A-11008 | Secondary antibody for IF |
| antibody | AlexaFluor-594 secondary antibody anti mouse | Life Technology | A-11005 | Secondary antibody for IF |
| antibody | AlexaFluor-594 secondary antibody anti rabbit | Life Technology | A-11012 | Secondary antibody for IF |
| antibody | mouse anti-CAP-D3 antibody | Santa Cruz | Sc-81597 | WB (1:1000) |
| antibody | goat DyLight 800 florescent anti-mouse secondary antibodies | Cell Signaling Technology | Cat #5,257 | WB (1:5000) |
| recombinant DNA reagent | Mouse Ncaph2 cDNA | Origene | MC200537 | Initial cDNA used for cloning in pRNA |
| recombinant DNA reagent | Mouse Mcph1 cDNA | Dharmacon | 5697978 | Initial cDNA used for cloning in pRNA |
| recombinant DNA reagent | Mouse SMC2 cDNA | Source Bioscience | IRAV p968E12162D | Initial cDNA used for cloning in pRNA |
| recombinant DNA reagent | pRNA-NCAPH2-GFP | *Houlard et al., 2015* DOI:10.1038/ncb3167 | | In vitro transcription of NCAPH2 mRNA for mouse oocyte injection |
| recombinant DNA reagent | pRNA-Mad2 | *Houlard et al., 2015* DOI:10.1038/ncb3167 | | In vitro transcription of Mad 2 mRNA for mouse oocyte injection |
| recombinant DNA reagent | pRNA-H2b-mCherry | *Houlard et al., 2015* DOI: 10.1038/ncb3167 | | In vitro transcription of H2b-mCherry mRNA for mouse oocyte injection |
| recombinant DNA reagent | pRNA-MCPH1 | This paper | | In vitro transcription of mouse MCPH1 mRNA for mouse oocyte injection |

| Reagent type (species) or resource | Designation | Source or reference | Identifiers | Additional information |
|---|---|---|---|---|
| recombinant DNA reagent | pRNA-MCPH1-Δ200 | This paper | | In vitro transcription of mouse MCPH1 deleted of the N-terminal 200aa mRNA for mouse oocyte injection |
| recombinant DNA reagent | pRNA-Fusion SMC2-NCAPH2 | This paper | | In vitro transcription of the fusion SMC2-NCAPH2 mRNA for mouse oocyte injection |
| recombinant DNA reagent | pRNA-TEV | *Houlard et al., 2015* DOI:10.1038/ncb3167 | | In vitro transcription of TEV mRNA for mouse oocyte injection |
| recombinant DNA reagent | pSptCas9(BB)–2A-Puro | Addgene | 62,988 | Cloning of the SgRNA |
| recombinant DNA reagent | pUC19 | NEB | | Cloning of the targeting construct for CRISPR/Cas9 targeting |
| Recombinant DNA reagent | pLIB | Jan-Michael Peters lab, IMP | Addgene plasmid # 80,610 | |
| Recombinant DNA reagent | pLIB MCPH1 | Genscript | | |
| Recombinant DNA reagent | pET His6 MBP TEV LIC cloning vector | Scott Gradia lab, UC Berkeley | Addgene plasmid # 29,656 | |
| Recombinant DNA reagent | pET His6 MBP MCPH1 1–435 | This study | | |
| Recombinant DNA reagent | pET His6 MBP MCPH1 1–195 | This study | | |
| Recombinant DNA reagent | pET His6 MBP MCPH1 196–435 | This study | | |
| Recombinant DNA reagent | pET His6 MCPH1 1–435 MBP | This study | | |
| sequence-based reagent | GGTGGAAAGTAGTATATACC | This paper | | Sg RNA used for: NCAPH2-GFP |
| sequence-based reagent | GGTGGAAAGTAGTATATACC | This paper | | Sg RNA used for: NCAPH2-Halo |
| sequence-based reagent | GGTGTGCAATTCCTAGTGTG | This paper | | Sg RNA used for: MCPH1 deletion Sg5' |
| sequence-based reagent | AGCTGTTCCTTAGAACACGA | This paper | | Sg RNA used for: MCPH1 deletion Sg3' |
| sequence-based reagent | ACAGTGAGACATCTACAATG | This paper | | Sg RNA used for: MCPH1-GFP |
| sequence-based reagent | CATGGATTTCTCCGGTGAAT | This paper | | Sg RNA used for: STOP-TEV |
| sequence-based reagent | AATGGGTGCTTATAATTAGC | This paper | | Sg RNA used for: WAPL-Lox-TEV Sg5' |
| sequence-based reagent | ACAATGTCACAATGGCTCAT | This paper | | Sg RNA used for: WAPL-Lox-TEV Sg3' |
| sequence-based reagent | ATAATATGGAACCGTGGTCC | This paper | | Sg RNA used for: SCC1-Halo |
| sequence-based reagent | cgttgaggcttcttcctatg | This paper | | Sg RNA used for: DELCEN-MCPH1 |
| sequence-based reagent | AATGGGTGCTTATAATTAGC and ACAATGTCACAATGGCTCAT | This paper | | Sg RNA used for: TEV sites in Wapl |
| peptide, recombinant protein | 5FAM-MCPH1$_{407-422}$ | Genscript | | |
| peptide, recombinant protein | MCPH1$_{407-422}$ | Genscript | | |
| peptide, recombinant protein | MCPH1$_{407-422}$pS417 | Genscript | | |

| Reagent type (species) or resource | Designation | Source or reference | Identifiers | Additional information |
|---|---|---|---|---|
| commercial assay or kit | Click-iT EdU Alexa Fluor 488 Imaging kit | In vitrogen | C10337 | EdU fluorescent labeling |
| commercial assay or kit | Gibson assembly | NEB | E5510S | Cloning kit |
| commercial assay or kit | GFP-trap agarose beads | Chromotek | GTA-10 | GFP tagged protein purification |
| Commercial assay or kit | HiTrap Q HP | cytiva | Cat #17–0407 | |
| Commercial assay or kit | HiTrap Heparin HP | cytiva | Cat #17–0407 | |
| Commercial assay or kit | StrepTrap HP | cytiva | Cat #28–9075 | |
| Commercial assay or kit | Superose 6 Increase 10/300 GL | cytiva | Cat #29-0915-96 | |
| Commercial assay or kit | Superdex 200 Increase 10/300 GL | cytiva | Cat # 28990944 | |
| Commercial assay or kit | HiLoad 16/60 Superdex 200 | GE Healthcare | Cat# GE28-9893-35 | |
| Commercial assay or kit | EnzChek phosphate assay kit | Invitrogen | Cat# E6 | |
| Commercial assay or kit | 4%–12% NuPAGE Bis-Tris gels | Invitrogen | Cat #NW04125 | |
| Commercial assay or kit | Color Prestained Protein Standard, Broad Range 11–245 kDa | NEB | Cat # P7719 | |
| Commercial assay or kit | Pierce Silver Stain Kit | Thermo Scientific | Cat # 24,612 | |
| Commercial assay or kit | InstantBlue Coomassie Protein Stain | Abcam | Cat #ab119211 | |
| Chemical compound, drug | cellfectin II | Gibco, Thermo Fisher | Cat #10362100 | |
| Chemical compound, drug | FBS | Gibco, Thermo fisher | Cat #10082139 | |
| Chemical compound, drug | Pierce Protease Inhibitor Tablets | Thermo Scientific | Cat #A32965 | |
| Chemical compound, drug | Benzonase | millipore | Cat #E1014 | |
| Chemical compound, drug | HisPur Ni-NTA Resin | Thermo Scientific | Cat #88,221 | |
| Chemical compound, drug | Strep-tactin Sepharose resin | Iba-lifesciences | Cat # 2-1201-002 | |
| chemical compound, drug | HALO-PROTAC | Promega | CS2072A01 | In vivo degradation of Halo-NCAPH2 |
| chemical compound, drug | Halo-TMR | Promega | G8251 | Halo tagged protein detection |
| chemical compound, drug | Halo-JFX554 | **Grimm et al., 2021** DOI:10.1021/jacsau.1c00006 | | Halo tag detection by fluorescence |
| chemical compound, drug | RO-3306 | Sigma | SML0569 | CDK1 inhibitor |
| chemical compound, drug | Lipofectamine 2000 | Thermofisher | 11668030 | E14 transfection reagent |
| chemical compound, drug | Q5 DNA polymerase | NEB | M0491L | PCR amplification |
| software, algorithm | CRISPOR | | http://crispor.tefor.net/crispor.py | Guide RNA design |
| software, algorithm | EasyFRAP | | http://easyfrap.vmnet.upatras.gr | FRAP data analysis |
| software, algorithm | Fiji 2.0.0-rc-49/1.52i | NIH Image | http://fiji.sc/ | Image analysis and quantification |

| Reagent type (species) or resource | Designation | Source or reference | Identifiers | Additional information |
|---|---|---|---|---|
| Software | Image studio | Li-Cor | | |

