## [Editor Report]

This paper will be of broad interest to scientists in the field of chromosome biology and has high clinical relevance. It reveals how the MCPH1 protein, that is frequently mutated in disorders of brain growth, prevents premature chromosome condensation. Diverse and complementary approaches are used to provide a compelling argument that provides insight into the mechanism by which chromosome organisation is temporally controlled.

---

## [Decision Letter]

**Decision letter after peer review:**

Thank you for submitting your article "MCPH1 inhibits condensin II during interphase by regulating its SMC2-kleisin interface" for consideration by *eLife*. Your article has been reviewed by 3 peer reviewers, including Adèle L Marston as Reviewing Editor and Reviewer #1, and the evaluation has been overseen by Kevin Struhl as the Senior Editor. The following individual involved in review of your submission has agreed to reveal their identity: Andrew J Wood (Reviewer #2).

Essential revisions:

1. Please provide quantification of phenotypes throughout the manuscript. Specific examples are:

(a) Figure 1B/C, 1E, 3D: The fraction of cells with prophase like chromosomes should be quantified.

(b) Figure 1—figure supplement 1. Quantification of the centromere clustering phenotype should be provided.

(c) Figures 8B and 9C, the phenotype descriptions lack quantification. Proportion of oocytes showing chromosome missegregation should be quantified.

(d) It is an important observation that fusion condensin II could fully restore the chromosome segregation defects. The authors should also show the quantification data.

(e) It is stated in the discussion that in the absence of MCPH1, condensin II organises chromosomal DNAs into chromatid-like structures during G1 and G2 but not S phase. This is an important conclusion and should be documented with quantification.

2. Several figures/figure legends would benefit from more detail:

a) Figure 4 and Figure 4 —figure supplement 1. A better annotation of the HI-C contact matrices and explanation should be given. In particular, the general reader will need greater direction as to what features of the maps lead to the following observation: "enhances the frequency of inter-compartment (A to 275 B) contacts as compared to intra-compartment contacts (A to A and B to B) (Figure 4 276 —figure supplement 1A and B). In Figure 4A, please explain what is the yellow and black squares. In Figure 4D the arrow needs to be explained.

(b) In Figure 5 and supplemental figure 5, Coomassie and silver staining gels should be labeled to indicate which bands are which proteins.

(c) Figure 6D: The H2Ax foci are barely visible in the figure.

(d) Figure 8E -> what do the dots and crosses on the boxplots represent? There is only a single cross on the control box despite 11 measurements being reported in the legend.

(e) In the text describing Figure 10 it is stated that Wapl deletion does not alter condensin's distribution in interphase cells. However when the localisation of NCAPH2 is compared between the WT and WAPL delete in panel A, specifically between cells with low H3S10P staining, there appear to be thread like structures visible in the WAPL mutant but not wildtype. Assuming the authors' interpretation is correct, perhaps a more representative image could be used here.

(f) In line 537, please show Figure 11 —figure supplement 1.

3. In experiments of mouse oocytes, it is unclear how much endogenous Mcph1 is expressed and how abundant the exogenous Mcph1 is. These expression levels should be shown at least in western blotting.

4. In page 17 line 422-427, the authors mention that Mcph1 cannot dissociate condensin II in meiosis I, therefore claim that Mcph1 prevents the "initial loading" of condensin II on chromosomes but has little effect on complexes already stably associated with chromosomes. However, they injected Mcph1 in GV stage that is corresponding to G2 phase, when Mcph1 should be already loaded on chromatin. It is also conceivable that in mitosis, condensin II is located to the axis of condensed mitotic chromosomes and it could be hard to be accessed by Mcph1. Such difference in accessibility could also be the cause of different consequence between G2 and mitotic behavior of condensin II. To clarify this point, it would be better to perform Mcph1 over expression experiments in synchronizing culture cells and test if pre-loaded condensin II is resistant to Mcph1 in late G2 phase.

5. If fusion condensin II is resistant to Mcph1 while it is fully functional to restore NCAPH2 depletion, the fusion expression itself should cause precocious condensation like seen in Mcph1 depleted cells. Is this the case?*Reviewer #1:*

During prophase, condensin II drives the formation of chromosome loops to promote chromosome condensation, but this process is limited in interphase. In this manuscript the authors address how condensin II activity is limited by MCPH1. They show that MCPH1 limits condensin II-dependent loop formation, and provide evidence that it does so by opening of condensin II at its SMC2-NCAPH2 interface to remove it from the DNA. This is reminiscent of the mechanism by which WAPL removes cohesin during prophase. These conclusions are based on experiments using a diverse range of approaches in complementary systems, including ES cell engineering, in vitro biochemistry, Hi-C, live cell imaging of mouse oocytes. This range of approaches is a major strength of the manuscript. Overall, the manuscript presents a convincing case that MCPH1 counteracts condensin activity, in a similar way to restriction of cohesin activity by WAPL. This is an important advance in chromosome biology. Furthermore, since MCPH1 is frequently mutated in microcephaly patients, this work has direct medical relevance.

*Reviewer #2:*

The activity of the two mammalian condensin complexes is restricted to M phase. Condensin I activity is restricted at least partly via its sequestration in the cytoplasm during interphase. In contrast, a sizeable pool of condensin II remains in the nucleus throughout interphase and hence additional mechanisms are required to prevent inappropriate loop extrusion activity.

Prior work has implicated the MCPH1 protein as a negative regulator of condensin II activity, but how it exerts this function was unclear. Elucidating how MCPH1 regulates condensin II has potential clinical relevance: MCPH1 is the most frequently mutated gene in human congenital microcephaly, and condensin II mutations, although much rarer, cause a related phenotype.

This paper confirms prior findings and makes significant progress towards elucidating the mechanism through which MCPH1 regulates condensin II. The authors show that MCPH1 prevents condensin II from binding to, and condensing, chromosomes during G1 and G2 but (intriguingly) not during S phase. Interphase condensin II activity in MCPH1 mutant cells is shown to be independent of CDK1 activity, and therefore distinct from activation mechanisms that operate during mitotic prophase. The authors identify an evolutionarily conserved short linear motif on MCPH1 that interacts with NCAPG2, and provide evidence that it can be phosphorylated via CDK1, tentatively suggesting a mechanism via which MCPH1-mediated inhibition of condensin II loading might be alleviated to facilitate loading during mitosis.

An elegant series of rescue experiments performed in mouse oocytes showed that MCPH1 overexpression prevents NCAPH2 binding to chromosomes during meiosis I, which in turn prevents chromosome segregation during anaphase. In this system, inhibition of NCAPH2 binding requires the N-terminal BRCT domain of MCPH1. The most interesting finding of the paper, in my view, is that the effect of MCPH1 deletion on NCAPH2 loading is prevented by a genetic fusion between SMC2 and the NCAPH2 N-terminus, implying that MCPH1 might prevent condensin II loading by regulating the opening of this subunit interface. There are potential parallels here with Wapl-mediated cohesin removal, which involves regulated opening of the equivalent interface.

A technical strength of the paper is the diverse combination of experimental approaches, including live and fixed cell imaging, in vitro biochemistry and genomics, used to characterise how MCPH1 interacts with condensin, and how this interaction influences chromosome structure. Where possible, the perturbation experiments have been performed in gene edited and non-transformed mammalian cells, avoiding the potentially confounding effects of protein overexpression and background chromosomal instability.

The paper opens up several interesting questions for future studies, e.g. how is condensin II activity restricted during S phase in MCPH1-deficient cells? Is condensin II able to avoid the protein roadblocks present on interphase chromosomes that are thought to constrain cohesin-mediated loop extrusion? More insight into this latter question might be gleaned from cell cycle resolved Hi-C experiments. It seems possible that the high proportion of S phase cells in the unsynchronised mESC cultures used in Figure 4 would have made it more challenging to identify signal from G1 and G2 cells, in which condensin II activity is unleashed by MCPH1 deletion.

Overall, the conclusions are justified in my opinion, and it is commendable that the authors exercise caution in their interpretations and offer alternative explanations where appropriate.*Reviewer #3:*

Houlard et al., revisited the mechanisms of Mcph1 to inhibit condensin II in mouse cells and in biochemical approaches. Previous studies have shown that patient cells carrying Mcph1 mutations or Mcph1 knockdown cells undergo premature chromosome condensation that depends on condensin II. In this manuscript, by using biochemical and cell biological approaches, the authors found that NCAPG2 subunit in condensin II is required for the association with Mcph1 and the central short linear motif (SLiM) of Mcph1 is responsible for the association. The central SLiM inhibits neither ATPase activity of condensin nor condensin II binding to chromatin, suggesting that SLiM inhibits the activity of condensin II other than ATPase.

Hi-C data is one of the major advances of this study from previous works. The authors show that Mcph1 deletion enhances in the frequency of long-range, intra-chromosomal contacts and inter-compartment (A to B) contacts. This is consistent with the compaction of individual chromosomes after MCPH1 deletion.

The authors also tested the function of Mcph1 in mouse oocytes and showed that overexpression of Mcph1 caused chromosome missegregation that is similar phenotype of previously reported NCAPH2 depletion. In addition, the authors claim that Mcph1 and Wapl, a cohesin removal factor, function in a similar way because closure of Smc2-NCAPH2 interface, which corresponds to Smc3-Scc1 interface in cohesin and is known as an exit gate opened by the action of Wapl, could bypass the condensin removal activity of Mcph1. Finally, in mouse embryonic stem cells, they performed double knockdown of Wapl and Mcph1 and showed that these factors act independently to each other. Intriguingly, Mcph1 knockdown transforms interphase cohesin axis formed by Wapl knockdown into solenoidal axis.

Their conclusions are mostly well supported by biochemical data, but several aspects of in vivo data need to be quantified

---

## [Author Response]

Essential revisions:1. Please provide quantification of phenotypes throughout the manuscript. Specific examples are:(a) Figure 1B/C, 1E, 3D: The fraction of cells with prophase like chromosomes should be quantified.

Figure 1B: the cell count was added in the main text: line 194:

“Immunofluorescence microscopy revealed that *Mcph1* deletion leads to a substantial increase in the fraction of cells with prophase like chromosomes: WT: 6.4%, 12/188 cells compared to ΔMcph1: 39.1%, 70/179 cells”

Figure 1C and 3D: the cell count was added in the main text line 200:

“Crucially, all EdU negative cells, whether they were in G1 (H3PS10 negative, 200 cells counted) or G2 (H3PS10 positive, 200 cells counted), contained prophase-like chromosomes while no condensation was observed in EdU positive S phase cells (200 cells counted) (Figure 1C and 3D).”

Figure 1E: the cell count was added in the main text line 225:

“Because this completely reversed the re-organisation of chromosomal DNA caused by *Mcph1* deletion (no condensation observed in 150 cells counted, Figure 1E), we conclude that altered regulation of condensin II is largely if not completely responsible.”

Figure 3D: the cell count was added in the main text line 262:

“The DNA organisation of wild type cells resembled that of normal G2 cells, namely no chromatid-like structures were formed and centromeres were clustered in chromocenters (200 cells counted, Figure 3D). In contrast, the chromosomal DNAs of all mutant cells were organized into prophase-like chromatids with individualised centromeres (150 cells counted, Figure 3D).”

(b) Figure 1—figure supplement 1. Quantification of the centromere clustering phenotype should be provided.

The quantification of the number of CREST foci per nucleus, their intensity and their size were added to Figure 1 —figure supplement 2. The counting was done over the entire population of cells irrespectively of their cell cycle stage as, even during replication, cells deleted for *Mcph1* seem to have disorganised chromocenters. The figure legend was modified accordingly, line 752.

(c) Figures 8B and 9C, the phenotype descriptions lack quantification. Proportion of oocytes showing chromosome missegregation should be quantified.

The figure legend was modified accordingly to include:

Line 939, Figure 8B:

“number of oocytes analysed in three independent experiments and showing segregation defects: control :0/12; +MCPH1: 18/19”

Line 962, Figure 9C:

“total number of oocytes showing chromosome segregation defects in three experiments: NCAPH2-GFP+MCPH1: 17/19, Fusion-GFP+MCPH1: 0/18”

(d) It is an important observation that fusion condensin II could fully restore the chromosome segregation defects. The authors should also show the quantification data.

The main text was modified accordingly line 455:

“17 out of 18 oocytes deleted for *Ncaph2* and injected with the fusion segregated their chromosome without any defect whereas all the *Ncaph2* deleted oocytes showed chromosome stretching (n=10)”

(e) It is stated in the discussion that in the absence of MCPH1, condensin II organises chromosomal DNAs into chromatid-like structures during G1 and G2 but not S phase. This is an important conclusion and should be documented with quantification.

Figure 1C and 3D: the cell count was added in the main text, line 200:

“Crucially, all EdU negative cells, whether they were in G1 (H3PS10 negative, 200 cells counted) or G2 (H3PS10 positive, 200 cells counted), contained prophase-like chromosomes while no condensation was observed in EdU positive S phase cells (200 cells counted) (Figure 1C and 3D).”

2. Several figures/figure legends would benefit from more detail:a) Figure 4 and Figure 4 —figure supplement 1. A better annotation of the HI-C contact matrices and explanation should be given. In particular, the general reader will need greater direction as to what features of the maps lead to the following observation: "enhances the frequency of inter-compartment (A to 275 B) contacts as compared to intra-compartment contacts (A to A and B to B) (Figure 4 276 —figure supplement 1A and B). In Figure 4A, please explain what is the yellow and black squares. In Figure 4D the arrow needs to be explained.

The legend of Figure 4 was modified to include description of the yellow and black squares and arrow: line 788.

The legend of Figure 4—figure supplement 1 was modified to include an explanation of the inter and intra- compartment contacts: line 826.

(b) In Figure 5 and supplemental figure 5, Coomassie and silver staining gels should be labeled to indicate which bands are which proteins.

Figure 5 was modified to indicate which bands are condensin complex subunits and different MBP/MCPH1 constructs are indicated with a *. The manuscript was modified accordingly: line 847.

(c) Figure 6D: The H2Ax foci are barely visible in the figure.

Figure 6D was modified to make the H2Ax more visible.

(d) Figure 8E -> what do the dots and crosses on the boxplots represent? There is only a single cross on the control box despite 11 measurements being reported in the legend.

The boxplots of both Figure 8 and 9 were corrected.

(e) In the text describing Figure 10 it is stated that Wapl deletion does not alter condensin's distribution in interphase cells. However when the localisation of NCAPH2 is compared between the WT and WAPL delete in panel A, specifically between cells with low H3S10P staining, there appear to be thread like structures visible in the WAPL mutant but not wildtype. Assuming the authors' interpretation is correct, perhaps a more representative image could be used here.

Due to the modification of the chromatin organization in *Wapl* deleted cells, it is true that it can be misleading when looking at NCAPH2-GFP staining that does not appear as even as in wild type cells. The choice of this particular field is motivated by the fact that it shows the different steps of chromosome formation from early G2 to metaphase. In order to address if NCAPH2-GFP was enriched on the chromosome axis of Δ*Wapl* cells, the fluorescence of SCC1-Halo (red) and NCAPH2 GFP (green) was quantified across the axis of three different structure (a, b and c) in four representative cells in early G2 (1), late G2 (2), prophase (3) and metaphase (4) (distinguished using the H3PS10 staining). The quantifications show that NCAPH2 is enriched on the chromosome axis only in prophase and metaphase but not in G2. These results have been included in a new Figure 10—figure supplement 2 referred to in the manuscript line 518.

(f) In line 537, please show Figure 11 —figure supplement 1.

The manuscript was modified accordingly.

3. In experiments of mouse oocytes, it is unclear how much endogenous Mcph1 is expressed and how abundant the exogenous Mcph1 is. These expression levels should be shown at least in western blotting.

Analysis by western blotting of the protein expression levels is not feasible, as it would need a considerable amount of oocytes to detect the protein. This experiment would need an equal amount of oocytes injected with mRNA and would require sacrifice a lot of animals as an average of 10 mature oocytes can be isolated per female.

However, the overexpression levels of the different constructs (MCPH1 or MCPH1 deleted of the N-terminus) were done in the same condition. The fact that the version of MCPH1 with a deletion of the N-terminal domain does not impede on the association of condensin II with DNA suggests that the results observed are specific and not due to any side effect triggered by the injection process. Furthermore, this result is consistent with other studies using different systems (such as Wood et al., JBC 2008 and Yamashita et al., JCB, 2011) which observed that the N-terminal domain is essential for MCPH1 repression of condensin II activity.

4. In page 17 line 422-427, the authors mention that Mcph1 cannot dissociate condensin II in meiosis I, therefore claim that Mcph1 prevents the "initial loading" of condensin II on chromosomes but has little effect on complexes already stably associated with chromosomes. However, they injected Mcph1 in GV stage that is corresponding to G2 phase, when Mcph1 should be already loaded on chromatin. It is also conceivable that in mitosis, condensin II is located to the axis of condensed mitotic chromosomes and it could be hard to be accessed by Mcph1. Such difference in accessibility could also be the cause of different consequence between G2 and mitotic behavior of condensin II. To clarify this point, it would be better to perform Mcph1 over expression experiments in synchronizing culture cells and test if pre-loaded condensin II is resistant to Mcph1 in late G2 phase.

The point made in the manuscript relates to the comparison between the activity of WAPL on cohesin and the activity of MCPH1 on condensin II: can MCPH1 remove condensin II from pre-formed chromosomes like WAPL does with cohesin? We assume that the reviewers meant condensin and not MCPH1 when they say “when Mcph1 should be already loaded on chromatin”.

In the oocyte experiments, NCAPH2-GFP produced from mRNA injection at the GV stage is loaded onto chromosomes (Figure 8D).

When Mcph1 mRNA is co-injected at the GV stage with NCAPH2-GFP, MCPH1 is able to prevent the association of NCAPH2-GFP with chromosomes.

However, when oocytes are first injected at the GV stage with mRNA coding for NCAPH2-GFP and Mad2, the oocytes after release are arrested in metaphase I and condensin II is all along the chromosome arms. These oocytes are then injected a second time with Mcph1 mRNA and followed by microscopy. However, condensin II remains on the chromosomes of these oocytes (data not shown). Some reduction of condensin II can eventually be detected 5 hours after injection of MCPH1, twice the half-life of condensin II on chromosomes as reported previously in the same condition (Houlard et al., 2015). This lead to our suggestion that, unlike WAPL with cohesin, MCPH1 cannot actively remove the preloaded condensin II from the chromosomes.

In mES cells, condensin II is only visible at the centromere during G2, not along the chromosome arms. While an inducible over expression of MCPH1 in G2 might be able to reduce condensin II accumulation at the centromere, we do not see how it would distinguish between de novo loading or removal.

5. If fusion condensin II is resistant to Mcph1 while it is fully functional to restore NCAPH2 depletion, the fusion expression itself should cause precocious condensation like seen in Mcph1 depleted cells. Is this the case?

This is an experiment that we tried in E14 cells. To avoid overexpression of the fusion we used a Tetracyclin inducible promoter to drive the expression of the fusion after targeting at one specific locus (Tigre) in E14 genome. Unfortunately, while the fusion was expressed at high levels, it remained in the cytoplasm of the cell during interphase preventing any analysis.

However, the nuclear localization of the fusion is not a problem in oocytes as, during meiosis, the chromosome condensation begins after the nuclear envelope break down (GVBD). During live cell imaging of oocytes injected with the fusion, no apparent modification of the condensation kinetic were observed.